# Test-Time Policy Adaptation for Enhanced Multi-Turn Interactions with LLMs

## Abstract

Large Language Models (LLMs) employ multi-turn interaction as a fundamental paradigm for completing complex tasks. However, their performance often degrades in extended interactions, as they are typically trained on static, single-turn data, which hinders their ability to adapt to real-time user feedback. To address this limitation, we first propose a new paradigm: _Test-Time Policy Adaptation for Multi-Turn Interactions_ ($T^2PAM$), which utilizes user feedback from the ongoing interaction as a reward signal to estimate a latent optimal policy aligned with user preferences, then updates a small subset of parameters to steer the model toward this policy, ultimately enabling efficient in-conversation self-correction. We then introduce _Optimum-Referenced One-Step Adaptation_ (ROSA), a lightweight algorithm that operationalizes $T^2PAM$. ROSA guides the model parameters toward a theoretical optimal policy in a single, efficient update step, avoiding costly iterative gradient-based optimization and minimizing computational overhead. We provide a rigorous theoretical analysis guaranteeing that the policy of ROSA converges to the preference of user as the number of interactions increases. Extensive experiments on challenging benchmark demonstrate that ROSA achieves significant improvements in both task effectiveness and efficiency.

## 1 Introduction

Multi-turn conversation is the predominant interaction paradigm between human and _Large Language Models_ (LLMs) (Li et al., 2025b; Yi et al., 2025). This conversational modality is essential for real-world applications (Zhang et al., 2025a), as it enables users to progressively refine initially underspecified intentions into concrete objectives (Herlihy et al., 2024; Zheng et al., 2023), engaging the model in a collaborative problem-solving process (Chen et al., 2023). However, a fundamental mismatch exists between this prevalent use case and existing LLM alignment methodologies (Laban et al., 2025; Van Miltenburg et al., 2025). Prevailing alignment methods, _Supervised Fine-Tuning_ (SFT) (Chung et al., 2024; Wei et al., 2025a; Lester et al., 2021) and _Reinforcement Learning from Human Feedback_ (RLHF) (Ouyang et al., 2022; Wei et al., 2025c), predominantly rely on single-turn data for both training (Shu et al., 2024) and evaluation (Chang et al., 2023). This paradigm misalignment not only limits the potential of the model in complex interactions (Irvine et al., 2023; Hendrycks et al., 2021c), but also creates a significant gap between its benchmark performance and its practical utility (Shinn et al., 2023; Wu et al., 2024). Consequently, while the combination of SFT for imparting extensive knowledge (Chu et al., 2025; Wei et al., 2025b) and RLHF for aligning with human preferences (Rafailov et al., 2023; Meng et al., 2024) endows models with strong single-turn capabilities (Zeng et al., 2024), these models often exhibit a pronounced degradation in performance during multi-turn interactions (Wang et al., 2024). In fact, previous work has highlighted that such models often perform poorly in multi-turn scenarios, resulting in diminished capabilities and increased instability (Laban et al., 2025). While multi-turn training strategies have been explored (Shi et al., 2025; Qu et al., 2024; Chen et al., 2025a), they are frequently hindered by the prohibitive costs of collecting high-quality data and training on long context sequences (Li et al., 2025b).

To address these challenges, we propose a new paradigm: _Test-Time Policy Adaptation for Multi-Turn Interactions_($T^2PAM$), shifting the existing static training paradigm to a flexible test-time adaption paradigm. Specifically, this paradigm requires using a model trained in a single-turn interaction to perform effective and efficient online policy adaptation during multi-turn reasoning. This paradigm utilizes conversational user feedback as a reward signal to refine its policy and align its

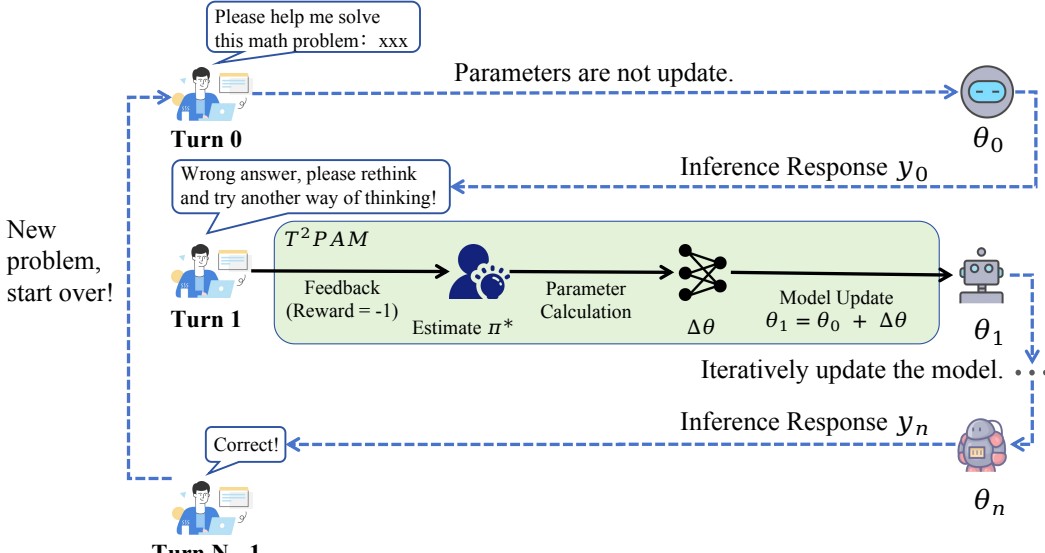

Figure 1: An illustration of the *Test-Time Policy Adaptation for Multi-Turn Interactions* (T$^2$PAM) paradigm. Different from static inference where the policy of model remains fixed ($\theta_0$, Turn 0), this paradigm treats conversational feedback as an active signal that guides real-time parameter updates (e.g., from $\theta_0$ to $\theta_1$). This iterative process of in-conversation self-correction allows the policy to progressively evolve and align with the preference of user ($\theta_n$) throughout the interaction.

behavior with the underlying intent of user, as illustrated in Figure 1. Importantly, this adaptation process must be computationally lightweight, so as to remain imperceptible to the user without incurring unaffordable inference latency or GPU memory overhead. Under this new paradigm, a model should be able to dynamically instantiate a user-specific policy for each conversational context, thereby enhancing the effectiveness and reliability of the multi-turn interaction.

Unfortunately, existing methodologies (Shani et al., 2024) are fundamentally misaligned with the requirements of T$^2$PAM. Specifically, (1) *Prompt Engineering* (Hu et al., 2024; Chen et al., 2023; Shinn et al., 2023) as a form of in-context learning, which adjusts the policy of model via contextual prompts, often fails to achieve effective preference alignment within a few interaction turns. (2) *Retrieval-Augmented Generation* (RAG) (Gao et al., 2024; Lewis et al., 2020), adapting the model output by lengthening the context, usually increases inference overhead significantly. Besides, its performance is determined by the quality and relevance of the external database. (3) *Model Editing* (ME) (Fang et al., 2025; Yao et al., 2023) is able to address the context length issue of RAG by internalizing knowledge as fact tuples through direct parameter updates. However, this representation is structurally unsuitable for encoding fine-grained user preferences. (4) Finally, existing *test-time methods* (Li et al., 2025a; Zuo et al., 2025; Hu et al., 2025; Liu et al., 2023) are primarily designed for single-turn tasks and often rely on extensive inference-time sampling. This process introduces significant computational costs and latency. Detailed related work is provided in the Appendix B.

To bridge this gap, we introduce *Optimum-Referenced One-Step Adaptation* (ROSA), a lightweight online adaptation algorithm that operationalizes our proposed paradigm T$^2$PAM. The core principle of ROSA is to leverage user feedback to analytically compute an estimate of the optimal policy and then steer the model towards this target in a single, efficient update step. This approach avoids costly iterative optimization, enabling principled in-conversation self-correction with minimal computational overhead. Our main contributions are summarized as follows:

- We demonstrate that current LLMs underperform in multi-turn interactions and propose T$^2$PAM paradigm to address this issue (Section 2).
- We propose ROSA, the first practical algorithm to implement this paradigm, which updates model parameters and align user preferences quickly during multi-turn interactions (Section 3).
- We establish a solid theory for ROSA, ensuring that its gap with user preferences narrows as the number of interaction turns increases (Section 4).
- We conduct extensive experiments on multiple challenging datasets. Our results show that ROSA outperforms baseline methods in both effectiveness and efficiency (Section 5).

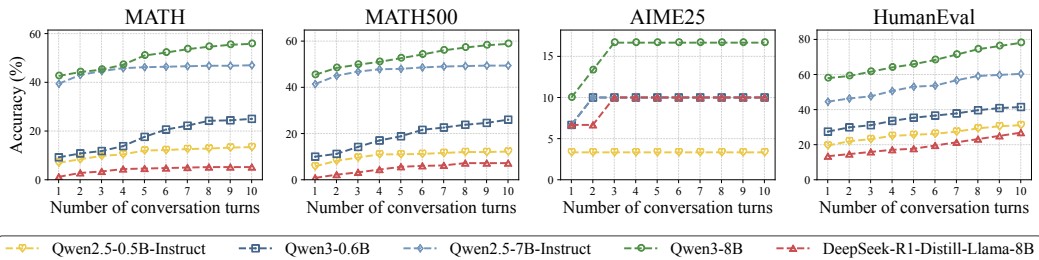

Figure 2: LLM accuracy after 10 rounds of interaction with humans. Although LLM accuracy shows a continuous and gradual improvement, this prompt-based correction process is inefficient.

## 2 THE T²PAM PARADIGM

The performance of LLMs often degrades in multi-turn interactions, because their alignment on static, single-turn datasets creates a paradigm mismatch that hinders their ability to adapt to user feedback or correct initial errors (Laban et al., 2025). To show this inefficiency, we empirically evaluated several LLMs on reasoning tasks. We first plot the cumulative accuracy over 10 conversational turns where human-like prompts were provided after each incorrect attempt. The results in Figure 2 show that while multi-turn interaction gradually improves accuracy, the process exhibits sharply diminishing returns. To diagnose this, Figure 3 plots the number of *newly* solved problems at each conversational

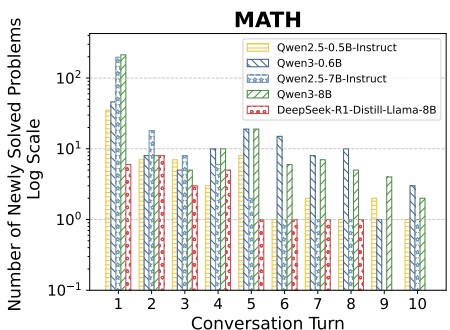

Figure 3: Number of *newly* solved problems per turn on the MATH dataset.

turn on the MATH dataset. The data reveal that the vast majority of problems are solved on the first attempt, with very few successful corrections in subsequent turns. This demonstrates that current models treat user interactions as passive context rather than as active signals for policy correction, highlighting a critical gap in their ability to perform efficient test-time adaptation.

To address this gap, we propose a new paradigm: *test-time policy adaptation for multi-turn interactions* (T²PAM). As summarized in Table 1, T²PAM resolves a trade-off faced by traditional approaches. While prompt-based lacks real-time adaptability and multi-turn training is costly and results in a static policy, T²PAM synthesizes the benefits of both. It operates during inference with zero training cost but, through online parameter modification, achieves high, policy-level adaptability that is more direct than prompting and more flexible than offline training. Notably, this paradigm shifts model alignment from a static, offline training stage to a dynamic, online inference process. More specifically, it requires methods that can update the policy of model in real-time by directly leveraging the rich feedback signals from a live conversation. We formally define T²PAM as below:

---

**Paradigm: Test-Time Policy Adaptation for Multi-Turn Interactions (T²PAM)**

Let a inference-time multi-turn interaction be a sequence of interactions indexed by turn $k \in \{1, \dots\}$. At the beginning of turn $k$, the language model is defined by a policy $\pi_{\theta_{k-1}}$ with parameters $\theta_{k-1}$. The paradigm proceeds as follows:

1. **Generation**: The model generates a response $\mathbf{y}_k \sim \pi_{\theta_{k-1}}(\cdot|\mathbf{x})$ given the conversational context $\mathbf{x}$.

2. **Feedback**: The subsequent interaction of user provides feedback, which is mapped to a scalar reward $r_k$ indicating task success (i.e., $r_k = +1$) or failure (i.e., $r_k = -1$).

3. **Adaptation**: If the task succeeds (i.e., $r_k = +1$), the multi-turn interaction is finished. Otherwise, an **effective and efficient** online adaptation function $\mathcal{A}$ updates the model parameters at inference time based on this failure feedback (i.e., $r_k = -1$) such that the model is more likely to succeed in the next turn:

$$\theta_k = \mathcal{A}(\theta_{k-1}, r_k, \mathbf{y}_k; \mathbf{x}) = \theta_{k-1} + \Delta\theta_k .$$

---

Table 1: Conceptual comparison of paradigms for improving multi-turn LLM performance.

| Feature | Prompt-based Methods | Multi-turn Data Training | T²PAM (Ours) |
|---|---|---|---|
| **Intervention Timing** | During inference | During training | During inference |
| **Operating Mode** | Reactive | **Proactive** (at training) | **Proactive** (at inference) |
| **Training Cost** | **Zero** (uses single-turn model) | High | **Zero** (uses single-turn model) |
| **Inference Cost** | Low (long context) | **Near-zero** | Low (preference alignment) |
| **Parameter Modification** | No | Yes (offline) | Yes (online) |
| **Real-time Adaptability** | Low (context-dependent) | Low (static policy) | **High** (policy-level) |

# 3 OPTIMUM-REFERENCED ONE-STEP ADAPTATION (ROSA)

To solve the paradigm we proposed above, we develop the *Optimum-Referenced One-Step Adaptation* (ROSA) approach (Algorithm 1), which enables effective and efficient online adaptation of a language model policy in direct response to real-time user feedback during multi-turn interactions. The core principle is to guide the model parameters towards a theoretical optimum in a single, efficient update step, avoiding iterative gradient-based optimization. This approach first defines the *Reinforcement Learning from Human Feedback* (RLHF) objective (Section 3.1) to maximize reward with KL regularization. It then leverages a closed-form analytical solution to directly identify the optimal policy (Section 3.2), applying exponential re-weighting to observed responses for practical one-step updates. Finally, parameter updates are efficiently computed via linearized optimization using the *Conjugate Gradient* algorithm (Section 3.3).

## 3.1 THE RLHF OBJECTIVE FOR TURN-WISE ADAPTATION

We propose to solve the T²PAM paradigm above using *Reinforcement Learning from Human Feedback* (RLHF) techniques (Ouyang et al., 2022). In this approach, we learn from a reward signal $r(\mathbf{x}, \mathbf{y})$ that reflects human preference given the context $\mathbf{x}$ and the response $\mathbf{y}$. Specifically, we model this feedback as a binary signal where $r(\mathbf{x}, \mathbf{y}) \in \{-1, +1\}$ corresponds to negative and positive feedback, respectively. The objective is to find an updated policy $\pi_\theta$ that maximizes the expected reward while penalizing significant divergence from the policy of the previous turn $\pi_{\theta_{k-1}}$ for stable and controlled updates. The deviation is measured by the *Kullback-Leibler* (KL) divergence. This leads to the following turn-wise optimization objective for turn $k$:

$$\max_{\pi_\theta} \quad \mathbb{E}_{\mathbf{y} \sim \pi_\theta(\cdot|\mathbf{x})} [r(\mathbf{x}, \mathbf{y})] - \beta D_{\mathrm{KL}} \left( \pi_\theta(\cdot|\mathbf{x}) \| \pi_{\theta_{k-1}}(\cdot|\mathbf{x}) \right) \tag{1}$$

where $\beta > 0$ is a coefficient that controls the strength of the KL regularization.

## 3.2 FROM THEORETICAL OPTIMUM TO A PRACTICAL ONE-STEP UPDATE

While the objective presented in (1) is conventionally optimized using iterative gradient-based methods (Sra et al., 2011; Kingma & Ba, 2017), such approaches are often characterized by their computational intensity and slow convergence, rendering them impractical for real-time online adaptation scenarios. Our methodology circumvents this inefficiency by leveraging a critical insight: this specific optimization problem admits a well-established closed-form analytical solution (Rafailov et al., 2023). Rather than relying on incremental approximations, we can directly ascertain the optimal policy. This foundational result is formalized in Theorem 1 (proof in Appendix C.1).

**Theorem 1** (Closed-Form Optimal Policy). *Let* $Z_k(\mathbf{x}) = \sum_{\mathbf{y}' \in \mathcal{Y}} \pi_{\theta_{k-1}}(\mathbf{y}'|\mathbf{x}) \exp\left(\frac{1}{\beta} r(\mathbf{x}, \mathbf{y}')\right)$ *be the partition function over the entire response space* $\mathcal{Y}$*, the policy* $\pi_{\theta_k}^*$ *that maximizes the turn-wise RLHF objective in* (1) *is given by:*

$$\pi_{\theta_k}^*(\mathbf{y}|\mathbf{x}) = \frac{1}{Z_k(\mathbf{x})} \pi_{\theta_{k-1}}(\mathbf{y}|\mathbf{x}) \exp\left(\frac{1}{\beta} r(\mathbf{x}, \mathbf{y})\right). \tag{2}$$

Theorem 1 demonstrates that the optimal policy is a re-weighted version of the reference policy, where the probability of a given response is exponentially modulated by its associated reward. In practical applications, feedback is typically received for only a *single* generated response, $\mathbf{y}_k$, often corresponding to a negative reward ($r_k = -1$) for an incorrect output. This constraint necessitates

---

**Algorithm 1** Optimum-Referenced One-Step Adaptation (ROSA)

---

1: **Input:** Initial model parameters $\theta_0$, hyperparameter $\beta$.
2: $k \leftarrow 1$
3: **while true do**
4:   // Step 1: Generate response and receive feedback
5:   Generate response $\mathbf{y}_k \sim \pi_{\theta_{k-1}}(\cdot|\mathbf{x})$.
6:   Receive reward $r_k$ based on user feedback.
7:   **if** $r_k = +1$ **then**
8:     **Terminate** // Stop immediately on success signal
9:   **end if**
10:   // Step 2: Construct the practical online target (Section 3.2)
11:   Compute target value $\tilde{\pi}^*_{\theta_k}(\mathbf{y}_k|\mathbf{x}) = \frac{1}{Z_k(\mathbf{x})}\pi_{\theta_{k-1}}(\mathbf{y}_k|\mathbf{x})\exp\left(\frac{1}{\beta}r_k\right)$.
12:   // Step 3: Compute parameter update via linearized optimization (Section 3.3)
13:   Define residual $\mathbf{d}_k = \tilde{\pi}^*_{\theta_k} - \pi_{\theta_{k-1}}$.
14:   Solve $(\mathbf{J}_k^\top \mathbf{J}_k)\Delta\theta_k = \mathbf{J}_k^\top \mathbf{d}_k$ for $\Delta\theta_k$ using *Conjugate Gradient* method.
15:   // Step 4: Update model parameters
16:   Update parameters: $\theta_k \leftarrow \theta_{k-1} + \Delta\theta_k$.
17: **end while**

---

the construction of an update target utilizing solely the observed data point $(\mathbf{x}, \mathbf{y}_k, r_k)$. We achieve this by applying the exponential re-weighting derived from the optimal policy in (2) exclusively to the observed response, thereby yielding a practical target value (derivation in Appendix C.2):

$$\tilde{\pi}^*_{\theta_k}(\mathbf{y}|\mathbf{x}) = \begin{cases} \frac{1}{Z_k(\mathbf{x})}\pi_{\theta_{k-1}}(\mathbf{y}|\mathbf{x})\exp\left(\frac{1}{\beta}r_k\right), & \text{if } \mathbf{y} = \mathbf{y}_k, \\ \frac{1}{Z_k(\mathbf{x})}\pi_{\theta_{k-1}}(\mathbf{y}|\mathbf{x}), & \text{if } \mathbf{y} \neq \mathbf{y}_k . \end{cases} \tag{3}$$

where $Z_k(\mathbf{x}) = 1 - \left(1 - \exp\left(\frac{1}{\beta}r_k\right)\right)\pi_{\theta_{k-1}}(\mathbf{y}_k|\mathbf{x})$. This formulation provides a direct learning signal for a one-step parameter update. For an incorrect response with reward $r_k = -1$, the target probability is scaled down relative to the current policy, effectively instructing the model to diminish the likelihood of generating that specific erroneous output in the future. This approach transforms an otherwise intractable global optimization problem into a targeted, sample-wise correction, forming the fundamental basis for our efficient adaptation mechanism.

### 3.3 EFFICIENT PARAMETER UPDATE VIA LINEARIZED OPTIMIZATION

With a practical target policy $\tilde{\pi}^*_{\theta_k}$ established, the subsequent step involves computing the parameter update $\Delta\theta_k$ that adjusts the current policy $\pi_{\theta_{k-1}}$ towards this target. This is accomplished through linearized optimization. This linearization is chosen for its computational ease and efficiency, allowing for rapid online adaptation without the prohibitive costs of higher-order optimization methods, as demonstrated in our efficiency analysis in Section E.3. Initially, the policy function is approximated using a first-order Taylor expansion around the current parameters $\theta_{k-1}$:

$$\pi_{\theta_{k-1}+\Delta\theta_k}(\mathbf{y}_k|\mathbf{x}) \approx \pi_{\theta_{k-1}}(\mathbf{y}_k|\mathbf{x}) + \nabla_\theta\pi_{\theta_{k-1}}(\mathbf{y}_k|\mathbf{x})^\top\Delta\theta_k . \tag{4}$$

Our objective is to determine $\Delta\theta_k$ such that the updated policy $\pi_{\theta_{k-1}+\Delta\theta_k}$ closely matches our target $\tilde{\pi}^*_{\theta_k}$. For the single data point $(\mathbf{x}, \mathbf{y}_k)$, this yields a linear system of equations:

$$\mathbf{J}_k\Delta\theta_k \approx \tilde{\pi}^*_{\theta_k}(\mathbf{y}_k|\mathbf{x}) - \pi_{\theta_{k-1}}(\mathbf{y}_k|\mathbf{x}) . \tag{5}$$

where $\mathbf{J}_k = \nabla_\theta\pi_{\theta_{k-1}}(\mathbf{y}_k|\mathbf{x})^\top$ represents the Jacobian of the policy output with respect to the model parameters. To obtain a stable, least-squares solution for $\Delta\theta_k$, we solve the following equations:

$$(\mathbf{J}_k^\top\mathbf{J}_k)\Delta\theta_k = \mathbf{J}_k^\top\left(\tilde{\pi}^*_{\theta_k}(\mathbf{y}_k|\mathbf{x}) - \pi_{\theta_{k-1}}(\mathbf{y}_k|\mathbf{x})\right) . \tag{6}$$

Explicitly forming the Hessian-approximating matrix $\mathbf{J}_k^\top\mathbf{J}_k$ is computationally prohibitive for models with a large number of parameters. As a consequence, we employ the *Conjugate Gradient (CG) algorithm* (Atkinson, 1988), an iterative solver that efficiently determines the solution to (6) without materializing this matrix. This is critical for memory efficiency, as it avoids storing the full Hessian-like matrix, making our approach incur less GPU memory overhead, as shown in Appendix E.3. The

CG algorithm only requires the computation of the matrix-vector product $(\mathbf{J}_k^\top \mathbf{J}_k)\mathbf{p}$ for an arbitrary vector $\mathbf{p}$. This computation is performed in a matrix-free manner by efficiently chaining two operations using automatic differentiation: a Jacobian-vector product (JVP) to compute $\mathbf{J}_k \mathbf{p}$, followed by a vector-Jacobian product (VJP) to compute $\mathbf{J}_k^\top (\mathbf{J}_k \mathbf{p})$.

Once the optimal $\Delta\theta_k$ is computed via CG method, the model parameters are updated in one step:

$$\theta_k \leftarrow \theta_{k-1} + \Delta\theta_k \ . \tag{7}$$

This entire procedure, encompassing feedback reception and parameter update computation, constitutes one complete cycle of ROSA, as comprehensively detailed in Algorithm 1.

## 4 THEORETICAL RESULTS

Having established the mechanics of ROSA, we now provide its theoretical underpinnings. This section demonstrates that our ROSA is not merely an effective heuristic but a principled algorithm with formal guarantees. Our analysis unfolds in three stages: we first prove that each corrective step is guaranteed to be productive (Section 4.1), then show that these gains accumulate over time to ensure convergence (Section 4.2), and finally, provide a unified bound that accounts for the practical approximation errors inherent in our efficient update step (Section 4.3).

Of note, a central aspect of our theoretical analysis revolves around the *Kullback-Leibler* (KL) divergence, specifically $D_{\text{KL}}(\pi_{\text{user}}^* \| \tilde{\pi}_{\theta_k}^*)$. This metric quantifies the dissimilarity between the underlying user optimal policy $\pi_{\text{user}}^*$ (representing the true preferences from the user and the ideal way to solve the task) and our adapted policy $\tilde{\pi}_{\theta_k}^*$. Minimizing this divergence is crucial because it directly implies that the generated responses from a model are becoming increasingly aligned with what the user desires and expects. When the model policy closely mirrors the user optimal policy, it is inherently more likely to produce correct and satisfactory outputs, thereby increasing the probability of task success and reducing the number of interaction turns required to achieve user intent.

### 4.1 MONOTONIC ERROR REDUCTION

Our first key result establishes that the adaptation mechanism in ROSA is *provably productive*. Each time the model receives corrective feedback, the resulting update is guaranteed to reduce the KL divergence between the underlying user policy and our estimated target policy, as formally shown in Theorem 2 (proof in Appendix C.3).

> **Theorem 2** (Monotonic Error Reduction). *Let $\pi_{\text{user}}^*$ be the underlying user policy and $\tilde{\pi}_{\theta_k}^*$ be the practical target policy in (3) after receiving feedback $r_k$ on response $\mathbf{y}_k$ at turn $k$. Suppose $\pi_{\theta_k} = \tilde{\pi}_{\theta_k}^*$ by applying exact policy update in ROSA, the change in KL divergence from the previous turn is bounded as follows:*
>
> $$D_{\text{KL}}(\pi_{\text{user}}^* \| \tilde{\pi}_{\theta_k}^*) - D_{\text{KL}}(\pi_{\text{user}}^* \| \tilde{\pi}_{\theta_{k-1}}^*) \leq -\frac{1}{\beta} \pi_{\text{user}}^*(\mathbf{y}_k|\mathbf{x}) \ . \tag{8}$$

**Remark.** This theorem provides a powerful guarantee for the reliability of ROSA. The most inspiring insight is that *every piece of corrective feedback is guaranteed to be productive*, confirming that learning from failure is a mathematically valid mechanism in our framework. The magnitude of this reduction is also highly informative. The term $\frac{1}{\beta}$ works as a learning rate; a smaller $\beta$ yields a more aggressive update, theoretically explaining the faster initial gains seen in our ablation study. Besides, the $\pi_{\text{user}}^*(\mathbf{y}_k|\mathbf{x})$ term reveals that the most impactful learning signals come from correcting plausible mistakes (high $\pi_{\text{user}}^*$ with $r = -1$) instead of nonsensical ones. Finally, this result provides strong theoretical justification for the one-step adaptation design in ROSA. As a single update is provably beneficial, the algorithm effectively avoids the complexity and potential instability of iterative optimization within a single turn.

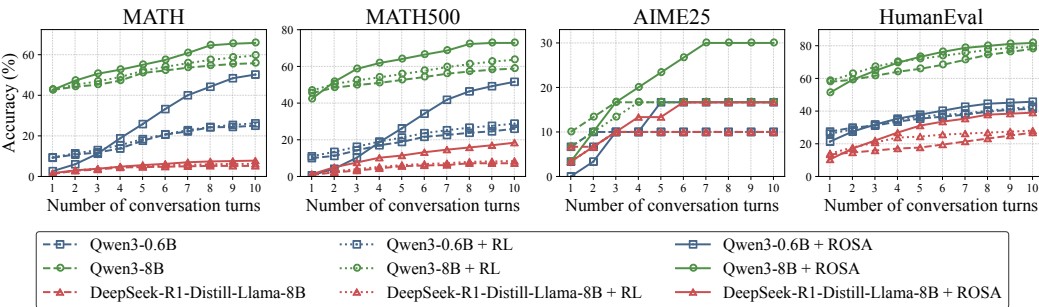

Figure 4: ROSA significantly boosts the rate of accuracy improvement in multi-turn interactions. These charts compare *baseline models*, *RL* described in Appendix E.4.1, and *ROSA* on different datasets. In contrast to the slow improvement shown in Figure 2, ROSA not only achieves a higher absolute accuracy but also accelerates the learning process, as evidenced by the steeper slopes of the solid lines. This highlights efficiency of ROSA in online error correction.

## 4.2 Cumulative Convergence Guarantee

While Theorem 2 guarantees improvement at each step, our second theorem extends this result to the entire multi-turn interaction, providing a bound on the cumulative error and ensuring **long-term convergence** in our Theorem 3 (proof in Appendix C.4).

**Theorem 3** (Cumulative Error Bound). *Suppose $\pi_{\theta_k} = \tilde{\pi}_{\theta_k}^*$ by applying exact policy update in ROSA, after $K$ turns of interaction, the KL divergence between the underlying user policy $\pi_{user}^*$ and the practical target policy $\tilde{\pi}_{\theta_K}^*$ in (3) is bounded as follows:*

$$D_{\mathrm{KL}}(\pi_{\mathrm{user}}^* \| \tilde{\pi}_{\theta_K}^*) \leq D_{\mathrm{KL}}(\pi_{\mathrm{user}}^* \| \pi_{\theta_0}) - \frac{1}{\beta} \sum_{k=1}^{K} \pi_{\mathrm{user}}^*(\mathbf{y}_k|\mathbf{x}) \,. \tag{9}$$

**Remark.** This theorem formalizes the core value proposition of multi-turn interaction within the ROSA framework. First, *the benefits of adaptation accumulate over time*. The summation term grows with each turn of feedback, progressively tightening the upper bound on the error. This formally demonstrates that the more a user interacts with the model, the closer the model policy will align with their true intent. Second, this result provides a *clear path to convergence*. As the number of turns $K$ increases, the cumulative subtracted term grows, forcing the error to decrease and ensuring the adaptation process is on a trajectory guaranteed to converge toward the optimal policy of user.

## 4.3 Unified Error Bound for the Adapted Policy

The previous theorems guarantee our *target* policy improves. However, the *final* policy, $\pi_{\theta_k}$, is subject to the approximation error from the first-order Taylor expansion used for our efficient update. The following unified theorem combines the guaranteed improvement from feedback with the accumulated linearization error to provide a comprehensive bound on the true performance of ROSA (proof in Appendix C.5).

**Theorem 4** (Unified Convergence Bound). *Assume $\log \pi_\theta$ is Lipschitz-smooth with constant $L$. After $K$ turns of interaction in ROSA, the divergence of the final adapted policy $\pi_{\theta_K}$ from the underlying user policy $\pi_{user}^*$ is bounded by:*

$$D_{\mathrm{KL}}(\pi_{\mathrm{user}}^* \| \pi_{\theta_K}) \leq \underbrace{D_{\mathrm{KL}}(\pi_{\mathrm{user}}^* \| \pi_{\theta_0})}_{\text{Initial Error}} - \underbrace{\frac{1}{\beta} \sum_{k=1}^{K} \pi_{\mathrm{user}}^*(\mathbf{y}_k|\mathbf{x})}_{\text{Improvement}} + \underbrace{\frac{L}{2} \sum_{k=1}^{K} \|\Delta\theta_k\|_2^2}_{\text{Approx. Error}} \,. \tag{10}$$

**Remark.** This unified bound rigorously quantifies the inherent trade-off in online policy adaptation. Each turn reduces the KL divergence from the underlying user optimal policy by a reward-driven

term $\frac{1}{\beta}\pi^*_{\text{user}}(\mathbf{y}_k|\mathbf{x})$, while incurring an approximation error $\frac{L}{2}\|\Delta\theta_k\|^2_2$ due to linearization. Convergence requires the net progress per turn to remain positive. This balance is affected by two factors. Firstly, the approximation error is controlled because $\pi_{\theta_{k-1}}(\mathbf{y}_k|\mathbf{x})$ is typically small in practice, limiting the magnitude of $\Delta\theta_k$ according to (3). This ensures the improvement from a potentially large $\pi^*_{\text{user}}(\mathbf{y}_k|\mathbf{x})$ can effectively outweigh the approximation cost. Secondly, the regularization coefficient $\beta$ modulates this trade-off: a smaller $\beta$ accelerates learning but risks amplifying approximation error, while a larger $\beta$ stabilizes updates at the cost of slower progress. This interplay explains the two-phase behavior observed in practice: rapid initial corrections followed by stable, fine-grained refinements, as detailed in Appendix E.4.2. The theorem therefore serves as both a robust theoretical guarantee and a practical design guide for balancing adaptation speed and stability.

Table 2: Main results of ROSA across diverse task domains, reporting accuracy (%). We compare the **Baseline** (standard multi-turn interaction) with several variants of ROSA. The notation '(+A+B)' indicates the update location (A: "LM" for LM Head, "HS" for Hidden States) and the reward model type (B: "R" for rule-based, "M" for model-based). The values in red denote the absolute improvement over the baseline. Further details on parameter updates and reward models are provided in Appendix D.5 and D.4, respectively.

| Model | Method | Mathematical Reasoning | | General Reasoning | | Multilingual Reasoning | | Code Gen. |
| | | MATH | MATH-500 | MMLU-R | SuperGPQA | MT-AIME24 | MT-MATH100 | HumanEval |
|---|---|---|---|---|---|---|---|---|
| Qwen2.5-0.5B -Instruct | Baseline | 13.40 | 12.20 | 7.27 | 1.90 | 3.48 | 15.40 | 31.09 |
| | ROSA (+LM + R) | 30.40 (+17.00) | 28.00 (+15.80) | 9.07 (+1.80) | 5.63 (+3.73) | 3.67 (+0.19) | 22.80 (+7.40) | 37.19 (+6.10) |
| | ROSA (+HS + R) | 25.40 (+12.00) | 25.00 (+12.80) | 11.00 (+3.73) | 5.00 (+3.10) | 4.90 (+1.42) | 20.90 (+5.50) | 37.27 (+6.18) |
| | ROSA (+LM + M) | 27.00 (+13.60) | 28.40 (+16.20) | 13.72 (+6.45) | 6.57 (+4.67) | 6.13 (+2.65) | 25.20 (+9.80) | 39.37 (+8.28) |
| Qwen3-0.6B | Baseline | 25.00 | 26.00 | 18.60 | 4.20 | 4.80 | 31.30 | 41.46 |
| | ROSA (+LM + R) | 50.20 (+25.20) | 51.60 (+25.60) | 33.40 (+14.80) | 9.13 (+4.93) | 7.58 (+2.78) | 56.60 (+25.30) | 45.73 (+4.27) |
| | ROSA (+HS + R) | 50.80 (+25.80) | 50.60 (+24.60) | 36.00 (+17.40) | 9.70 (+5.50) | 7.90 (+3.10) | 51.90 (+20.60) | 47.27 (+5.81) |
| | ROSA (+LM + M) | 52.20 (+27.20) | 54.60 (+28.60) | 40.68 (+22.08) | 15.73 (+11.53) | 9.43 (+4.63) | 59.40 (+28.10) | 49.37 (+7.91) |
| Qwen2.5-7B -Instruct | Baseline | 47.00 | 49.40 | 45.36 | 19.31 | 19.24 | 60.34 | 57.92 |
| | ROSA (+LM + R) | 63.40 (+16.40) | 62.40 (+13.00) | 62.17 (+16.81) | 37.26 (+17.95) | 27.14 (+7.90) | 73.16 (+12.82) | 63.41 (+5.49) |
| | ROSA (+HS + R) | 64.40 (+17.40) | 63.40 (+14.00) | 67.31 (+21.95) | 36.27 (+16.96) | 26.75 (+7.51) | 72.27 (+11.93) | 64.24 (+6.32) |
| | ROSA (+LM + M) | 65.20 (+18.20) | 65.60 (+16.20) | 68.47 (+23.11) | 40.67 (+21.36) | 30.21 (+10.97) | 75.13 (+14.79) | 67.36 (+9.44) |
| Qwen3-8B | Baseline | 55.80 | 58.80 | 51.35 | 27.61 | 30.37 | 74.74 | 78.04 |
| | ROSA (+LM + R) | 65.80 (+10.00) | 72.80 (+14.00) | 67.27 (+15.92) | 36.11 (+8.50) | 40.16 (+9.79) | 85.16 (+10.42) | 81.71 (+3.67) |
| | ROSA (+HS + R) | 65.80 (+10.00) | 66.20 (+7.40) | 68.37 (+17.02) | 37.73 (+10.12) | 42.27 (+11.90) | 86.93 (+12.19) | 82.37 (+4.33) |
| | ROSA (+LM + M) | 67.40 (+11.60) | 68.40 (+9.60) | 70.36 (+19.01) | 40.34 (+12.73) | 43.93 (+13.56) | 88.37 (+13.63) | 83.65 (+5.61) |
| DeepSeek-R1 -Distill-Llama-8B | Baseline | 5.20 | 7.20 | 30.46 | 10.37 | 4.73 | 17.35 | 25.00 |
| | ROSA (+LM + R) | 7.80 (+2.60) | 18.40 (+11.20) | 41.14 (+10.68) | 20.49 (+10.12) | 6.13 (+1.40) | 21.17 (+3.82) | 39.03 (+14.03) |
| | ROSA (+HS + R) | 8.40 (+3.20) | 18.20 (+11.00) | 42.18 (+11.72) | 21.34 (+10.97) | 7.27 (+2.54) | 23.85 (+6.50) | 38.37 (+13.37) |
| | ROSA (+LM + M) | 8.60 (+3.40) | 20.80 (+13.60) | 45.79 (+15.33) | 24.97 (+14.60) | 8.19 (+3.46) | 24.67 (+7.32) | 39.26 (+14.26) |

## 5  EMPIRICAL RESULTS

We conduct extensive experiments to validate the effectiveness and efficiency of our proposed ROSA framework in dynamic, multi-turn settings. In this section, we present our two primary findings: we first demonstrate the state-of-the-art performance of ROSA across a diverse range of tasks (Section 5.1), and then we analyze its effectiveness in online error correction (Section 5.2). A comprehensive description of our experimental setup, including the datasets, baselines, evaluation metrics, and reward models, is deferred to Appendix D. Furthermore, in-depth ablation studies analyzing our optimization strategy and the hyperparameter $\beta$ are provided in Appendix E.4.

### 5.1  EFFECTIVENESS AND GENERALIZABILITY ACROSS TASK DOMAINS

To validate the generalization ability and flexibility of ROSA, we first evaluated its performance across four different domains: mathematical reasoning, general reasoning, code generation, and multilingual reasoning. Detailed information about the datasets is provided in Appendix D.1. The results are shown in Table 2, and for more data sets and model results, see Appendix E.1. From the results, we draw several key conclusions. First, ROSA consistently outperforms the *baseline* method (standard multi-turn interaction) across all benchmark datasets and with different LLM models, demonstrating its broad **applicability** and **effectiveness**. Second, ROSA is highly **flexible**. It performs well regardless of whether the *LM Head* or *Hidden States* are updated (see Appendix D.5 for details on parameter updates), indicating its adaptability to different parameter update strategies. Furthermore, the also results highlight the impact of feedback granularity. The dense *model-based* reward, which provides fine-grained feedback on the reasoning process, consistently yields the best or near-best performance across almost all settings. This demonstrates that ROSA can effectively

Table 3: Comparison of Correction Uplift (%) on mathematical reasoning datasets.

| Model | Method | MATH | AIME25 | MATH-500 | HumanEval |
|---|---|---|---|---|---|
| Qwen2.5-0.5B-Instruct | Baseline | 6.88 | 0.00 | 6.79 | 14.39 |
| | ROSA | **25.48** (+18.60) | **6.67** (+6.67) | **24.05** (+17.26) | **26.09** (+11.70) |
| Qwen3-0.6B | Baseline | 17.40 | 3.57 | 17.78 | 19.33 |
| | ROSA | **48.87** (+31.47) | **16.67** (+13.10) | **51.31** (+33.53) | **31.01** (+11.68) |
| Qwen2.5-7B-Instruct | Baseline | 12.54 | 3.57 | 13.65 | 28.57 |
| | ROSA | **41.53** (+28.99) | **20.69** (+17.12) | **36.91** (+23.26) | **40.00** (+11.43) |
| Qwen3-8B | Baseline | 23.00 | 7.41 | 24.54 | 47.83 |
| | ROSA | **40.42** (+17.42) | **27.59** (+20.18) | **52.94** (+28.40) | **62.50** (+14.67) |
| DeepSeek-R1-Distill -Llama-8B | Baseline | 4.05 | 3.57 | 6.45 | 15.49 |
| | ROSA | **6.30** (+2.25) | **13.79** (+10.22) | **17.41** (+10.96) | **31.97** (+16.48) |

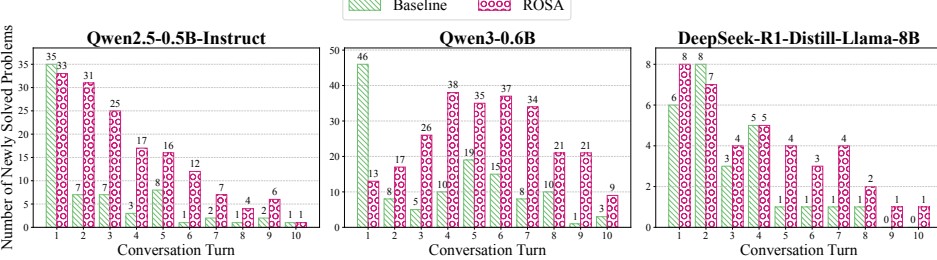

Figure 5: Comparison of newly solved problems per round on MATH datasets.

leverage detailed preference information to achieve superior alignment. On the contrary, we note that even with the sparser, *rule-based* reward, ROSA still delivers substantial improvements. This observation is consistent with our theoretical analysis in Theorem 2, which guarantees convergence even with simpler feedback signals. In addition, ROSA performance can reach or even outperform the multi turn training method (Appendix E.2).

Finally, we analyze the computational overhead of ROSA. Our results demonstrate that the method achieves its performance gains without a significant increase in inference latency or GPU memory consumption, enabling user-imperceptible policy optimization. The detailed inference time and memory usage metrics are provided in Appendix E.3. This efficiency is crucial, confirming that ROSA is a practical approach for enhancing multi-turn capabilities without additional overhead.

## 5.2 EFFECTIVENESS IN ONLINE ERROR CORRECTION

A core claim of our work is that ROSA enhances not just final accuracy, but the capacity of model for in-conversation self-correction. To quantify this, we propose the *Correction Uplift* metric, which measures the percentage of initially incorrect problems that are successfully solved in subsequent turns (see Appendix D.3 for details). The results in Table 3 show that ROSA dramatically improves this metric across all benchmarks, confirming its strong self-correction capability. This is further corroborated by the learning dynamics shown in our figures. In Figure 4, the accuracy curve for ROSA (solid line) exhibits a much steeper slope than the baselines, indicating a significantly faster rate of learning and correction. Figure 5 provides a more granular view: while the baseline model (green) shows sharply diminishing returns after the first turn, ROSA (purple) sustains a high rate of problem-solving in all subsequent rounds. This empirical result aligns with our theoretical analysis (Theorem 3), which establishes that ROSA learns from failures, enabling it to progressively align with user preferences. This capability is particularly impactful for small-scale LLM, substantially boosting their multi-turn reasoning performance. A detailed case study is provided in Appendix F.

## 6 CONCLUSIONS AND LIMITATIONS

In this work, we address the degradation of LLM performance in multi-turn dialogues by proposing a new paradigm $T^2PAM$, and its first practical implementation ROSA. ROSA enables efficient, in-conversation self-correction by updating model parameters online using real-time feedback. While our theoretical and experimental results validate ROSA, we acknowledge limitation that ROSA effectiveness is less effective on tasks that are heavily dependent on the model pre-trained knowledge.

ETHICS STATEMENT

We have manually reevaluated the dataset we created to ensure it is free of any potential for discrimination, human rights violations, bias, exploitation, and any other ethical concerns.

REPRODUCIBILITY STATEMENT

To ensure the reproducibility of our findings, all source code and datasets used in our experiments are included in the supplementary material. The provided materials are sufficient to replicate the main results presented in this paper.

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

## A  USAGE OF LLMS

Throughout the preparation of this manuscript, Large Language Models (LLMs) were utilized as a writing and editing tool. Specifically, we employed LLMs to improve the clarity and readability of the text, refine sentence structures, and correct grammatical errors. All final content, including the core scientific claims, experimental design, and conclusions, was conceived and written by us, and we take full responsibility for the final version of this paper.

## B  RELATED WORK

Research on improving the multi-turn capabilities of LLMs has largely proceeded along three main fronts: in-context learning, fine-tuning with multi-turn data, and reinforcement learning.

**In-Context Learning and Prompting Strategies.**  A prominent line of work enhances multi-turn performance without modifying model parameters by leveraging the context window to guide the model's reasoning (Ou et al., 2024; Sun et al., 2024). For instance, ChatCoT (Chen et al., 2023) models the chain-of-thought process as a multi-turn interaction to improve reasoning. Similarly, Reflexion (Shinn et al., 2023) refines model behavior by converting environmental feedback into textual summaries, which are appended to the context for subsequent turns. MathChat (Wu et al., 2024) extends this by introducing a user agent that can execute tools and inject the resulting feedback into the conversation. While effective, these methods are fundamentally limited by the model's intrinsic ability to interpret the provided context, and their performance is highly sensitive to the prompt design, which may even degrade performance in complex multi-turn scenarios if not perfectly aligned with the task.

**Fine-Tuning with Multi-Turn Data.**  Another approach involves fine-tuning the model on datasets specifically designed to capture multi-turn dynamics (Wang et al., 2024; Zhou et al., 2024). For instance, WildChat (Shi et al., 2025) leverages live user feedback to automatically construct a preference dataset for subsequent fine-tuning. Addressing challenges within this domain, Codesteer (Chen et al., 2025b) identifies a "gradient cancellation" issue, where gradients from early turns can interfere with those from later, more informative ones, and mitigates this by up-weighting the loss from the final turns of the interaction. However, a key limitation of such offline SFT approaches is their potential insufficiency in cultivating robust self-correcting behavior (Li et al., 2025b; Yi et al., 2025). This challenge often stems from a distribution mismatch between the errors present in the training data and those produced by the model at inference time, as well as the risk of "behavioral collapse," where the model overfits to a narrow set of correction patterns.

**Reinforcement Learning Approaches.**  Several methods employ reinforcement learning (RL) to teach models to self-improve over multiple rounds (Zhou et al., 2025; Liu et al., 2025). For instance, RISE (Qu et al., 2024) utilizes multi-round offline RL with reward supervision, applying a majority vote over candidate outputs at inference time. SCoRe (Kumar et al., 2024) adopts a two-stage process, first teaching the model to self-correct and then maximizing this capability via RL. Other works have explored multi-round group preference optimization by decomposing conversations into single-turn problems (Shi et al., 2024; Yang et al., 2025b). While these RL-based strategies can cultivate sophisticated, self-correcting behaviors, they often face significant challenges, including high computational costs and training instability, particularly when applied to long, multi-turn dialogue contexts.

While existing methods have advanced multi-turn capabilities, they present a fundamental trade-off. Offline approaches, such as fine-tuning and reinforcement learning, incur prohibitive computational costs associated with training on long contexts. Conversely, online in-context methods, while lightweight, are often inefficient at correcting a model's flawed intrinsic policy. Inspired by recent advances in test-time optimization (Zhang et al., 2025b; Zuo et al., 2025; Chen et al., 2025a), our work charts a new course. We introduce a novel paradigm, $T^2PAM$, that enables efficient, online policy modification *during inference*. This approach achieves the benefits of direct policy correction without the high cost of offline training and with greater flexibility than pure prompting strategies. We then present ROSA as the first practical algorithm to realize this paradigm.

## C PROOFS

### C.1 PROOF OF THEOREM 1

*Proof.* The policy $\pi_{\theta_k}^*$ that maximizes the turn-wise RLHF objective is found by reformulating the objective as a minimization problem. We begin with the objective from Equation 1 and combine terms inside the expectation:

$$J(\pi_\theta) = \max_{\pi_\theta} \quad \mathbb{E}_{\mathbf{y} \sim \pi_\theta(\cdot|\mathbf{x})} \left[r(\mathbf{x}, \mathbf{y})\right] - \beta D_{\text{KL}} \left(\pi_\theta(\cdot|\mathbf{x}) \,\|\, \pi_{\theta_{k-1}}(\cdot|\mathbf{x})\right) \tag{11}$$

$$= \max_{\pi_\theta} \mathbb{E}_{\mathbf{y} \sim \pi_\theta(\cdot|\mathbf{x})} \left[r(\mathbf{x}, \mathbf{y}) - \beta \log \left(\frac{\pi_\theta(\mathbf{y}|\mathbf{x})}{\pi_{\theta_{k-1}}(\mathbf{y}|\mathbf{x})}\right)\right] \tag{12}$$

Maximizing the above is equivalent to minimizing the negative of the term inside the expectation:

$$L(\pi_\theta) = \min_{\pi_\theta} \mathbb{E}_{\mathbf{y} \sim \pi_\theta(\cdot|\mathbf{x})} \left[\beta \log \left(\frac{\pi_\theta(\mathbf{y}|\mathbf{x})}{\pi_{\theta_{k-1}}(\mathbf{y}|\mathbf{x})}\right) - r(\mathbf{x}, \mathbf{y})\right] \tag{13}$$

$$= \min_{\pi_\theta} \mathbb{E}_{\mathbf{y} \sim \pi_\theta(\cdot|\mathbf{x})} \left[\log \left(\frac{\pi_\theta(\mathbf{y}|\mathbf{x})}{\pi_{\theta_{k-1}}(\mathbf{y}|\mathbf{x}) \exp(\frac{1}{\beta} r(\mathbf{x}, \mathbf{y}))}\right)\right] \tag{14}$$

We can recognize the denominator as being proportional to the optimal policy. Let us define the optimal policy $\pi_{\theta_k}^*$ by normalizing this term with the partition function $Z_k(\mathbf{x})$:

$$\pi_{\theta_k}^*(\mathbf{y}|\mathbf{x}) \triangleq \frac{1}{Z_k(\mathbf{x})} \pi_{\theta_{k-1}}(\mathbf{y}|\mathbf{x}) \exp \left(\frac{1}{\beta} r(\mathbf{x}, \mathbf{y})\right) \tag{15}$$

Substituting this definition back into the objective function:

$$L(\pi_\theta) = \min_{\pi_\theta} \mathbb{E}_{\mathbf{y} \sim \pi_\theta(\cdot|\mathbf{x})} \left[\log \left(\frac{\pi_\theta(\mathbf{y}|\mathbf{x})}{\pi_{\theta_k}^*(\mathbf{y}|\mathbf{x}) \cdot Z_k(\mathbf{x})}\right)\right] \tag{16}$$

$$= \min_{\pi_\theta} \left(\mathbb{E}_{\mathbf{y} \sim \pi_\theta(\cdot|\mathbf{x})} \left[\log \left(\frac{\pi_\theta(\mathbf{y}|\mathbf{x})}{\pi_{\theta_k}^*(\mathbf{y}|\mathbf{x})}\right)\right] - \mathbb{E}_{\mathbf{x}}[\log Z_k(\mathbf{x})]\right) \tag{17}$$

$$= \min_{\pi_\theta} \mathbb{E}_{\mathbf{y} \sim \pi_\theta(\cdot|\mathbf{x})} \left[\log \left(\frac{\pi_\theta(\mathbf{y}|\mathbf{x})}{\pi_{\theta_k}^*(\mathbf{y}|\mathbf{x})}\right)\right] \tag{18}$$

Since the partition function $Z_k(\mathbf{x})$ and its logarithm do not depend on the parameters of the policy $\pi_\theta$ being optimized, minimizing $L(\pi_\theta)$ is equivalent to minimizing the KL divergence between $\pi_\theta$ and the target optimal policy $\pi_{\theta_k}^*$:

$$\min_{\pi_\theta} \left[D_{\text{KL}}(\pi_\theta(\cdot|\mathbf{x}) \| \pi_{\theta_k}^*(\cdot|\mathbf{x}))\right] \tag{19}$$

The minimum value of the KL divergence is 0, which is achieved if and only if the two distributions are identical, i.e., $\pi_\theta = \pi_{\theta_k}^*$:

$$\pi_\theta(\mathbf{y}|\mathbf{x}) = \pi_{\theta_k}^*(\mathbf{y}|\mathbf{x}) = \frac{1}{Z_k(\mathbf{x})} \pi_{\theta_{k-1}}(\mathbf{y}|\mathbf{x}) \exp \left(\frac{1}{\beta} r(\mathbf{x}, \mathbf{y})\right) . \tag{20}$$

This completes the proof. $\qquad\square$

### C.2 DERIVATION OF EQUATION 3

**Definition 1** (Single-Sample Feedback Constraint). *In practical applications, feedback is typically received for only a single generated response, $\mathbf{y}_k$. We model this by constraining the general reward function $r(\mathbf{x}, \mathbf{y})$ as follows:*

$$r(\mathbf{x}, \mathbf{y}) = r_k \cdot \mathbb{I}(\mathbf{y} = \mathbf{y}_k) = \begin{cases} r_k, & \text{if } \mathbf{y} = \mathbf{y}_k \\ 0, & \text{if } \mathbf{y} \neq \mathbf{y}_k \end{cases} \tag{21}$$

*Derivation of the Practical Target from the Theoretical Optimum.* Our goal is to derive the practical, single-sample update target (Equation 3) and its corresponding partition function from the general theoretical optimal policy (Equation 2) under the Single-Sample Feedback Constraint (Definition 1).

**1. Derivation of the Practical Target Policy** $\tilde{\pi}^*_{\theta_k}$**.** We substitute the constrained reward from Assumption 1 into the general policy formula from Equation 2. This naturally yields a piecewise expression:

- For the observed response where $\mathbf{y} = \mathbf{y}_k$, the reward is $r_k$, yielding:

$$\tilde{\pi}^*_{\theta_k}(\mathbf{y}|\mathbf{x}) = \frac{1}{Z_k(\mathbf{x})}\pi_{\theta_{k-1}}(\mathbf{y}|\mathbf{x})\exp\left(\tfrac{1}{\beta}r_k\right) \tag{22}$$

- For all other responses where $\mathbf{y} \neq \mathbf{y}_k$, the reward is $0$, yielding:

$$\tilde{\pi}^*_{\theta_k}(\mathbf{y}|\mathbf{x}) = \frac{1}{Z_k(\mathbf{x})}\pi_{\theta_{k-1}}(\mathbf{y}|\mathbf{x})\exp(0) = \frac{1}{Z_k(\mathbf{x})}\pi_{\theta_{k-1}}(\mathbf{y}|\mathbf{x}) \tag{23}$$

Combining these two results gives the piecewise form in Equation 3.

**2. Derivation of the Practical Partition Function** $Z_k(\mathbf{x})$**.** Next, we apply the same constrained reward to the general partition function definition by splitting the sum over the entire response space $\mathcal{Y}$:

$$Z_k(\mathbf{x}) = \sum_{\mathbf{y}'\in\mathcal{Y}}\pi_{\theta_{k-1}}(\mathbf{y}'|\mathbf{x})\exp\left(\frac{1}{\beta}r_k\cdot\mathbb{I}(\mathbf{y}'=\mathbf{y}_k)\right)$$

$$= \pi_{\theta_{k-1}}(\mathbf{y}_k|\mathbf{x})\exp\left(\frac{1}{\beta}r_k\right) + \sum_{\mathbf{y}'\neq\mathbf{y}_k}\pi_{\theta_{k-1}}(\mathbf{y}'|\mathbf{x})\exp(0)$$

$$= \pi_{\theta_{k-1}}(\mathbf{y}_k|\mathbf{x})\exp\left(\frac{1}{\beta}r_k\right) + \left(1 - \pi_{\theta_{k-1}}(\mathbf{y}_k|\mathbf{x})\right)$$

$$= 1 - \left(1 - \exp\left(\frac{1}{\beta}r_k\right)\right)\pi_{\theta_{k-1}}(\mathbf{y}_k|\mathbf{x})$$

This confirms the expression for the practical partition function used in Equation 3. $\qquad\square$

### C.3 Proof of Theorem 2

*Proof.* We analyze the one-step change in error, $D_{\mathrm{KL}}(\pi^*_{\mathrm{user}}\|\pi^*_{\theta_k}) - D_{\mathrm{KL}}(\pi^*_{\mathrm{user}}\|\pi^*_{\theta_{k-1}})$.

$$D_{\mathrm{KL}}(\pi^*_{\mathrm{user}}\|\pi^*_{\theta_k}) - D_{\mathrm{KL}}(\pi^*_{\mathrm{user}}\|\pi^*_{\theta_{k-1}}) \tag{24}$$

$$= \left[\sum_{\mathbf{y}}\pi^*_{\mathrm{user}}(\mathbf{y})\log\left(\frac{\pi^*_{\mathrm{user}}(\mathbf{y})}{\pi^*_{\theta_k}(\mathbf{y})}\right)\right] - \left[\sum_{\mathbf{y}}\pi^*_{\mathrm{user}}(\mathbf{y})\log\left(\frac{\pi^*_{\mathrm{user}}(\mathbf{y})}{\pi^*_{\theta_{k-1}}(\mathbf{y})}\right)\right] \tag{25}$$

$$= \sum_{\mathbf{y}}\pi^*_{\mathrm{user}}(\mathbf{y})\left[\log\left(\frac{\pi^*_{\mathrm{user}}(\mathbf{y})}{\pi^*_{\theta_k}(\mathbf{y})}\right) - \log\left(\frac{\pi^*_{\mathrm{user}}(\mathbf{y})}{\pi^*_{\theta_{k-1}}}\right)\right] \tag{26}$$

$$= \sum_{\mathbf{y}}\pi^*_{\mathrm{user}}(\mathbf{y})\log\left(\frac{\frac{\pi^*_{\mathrm{user}}(\mathbf{y})}{\pi^*_{\theta_k}(\mathbf{y})}}{\frac{\pi^*_{\mathrm{user}}(\mathbf{y})}{\pi^*_{\theta_{k-1}}(\mathbf{y})}}\right) \tag{27}$$

$$= \sum_{\mathbf{y}}\pi^*_{\mathrm{user}}(\mathbf{y})\log\left(\frac{\pi^*_{\mathrm{user}}(\mathbf{y})}{\pi^*_{\theta_k}(\mathbf{y})}\cdot\frac{\pi^*_{\theta_{k-1}}(\mathbf{y})}{\pi^*_{\mathrm{user}}(\mathbf{y})}\right) \tag{28}$$

$$= \sum_{\mathbf{y}}\pi^*_{\mathrm{user}}(\mathbf{y})\log\left(\frac{\pi^*_{\theta_{k-1}}(\mathbf{y})}{\pi^*_{\theta_k}(\mathbf{y})}\right) \tag{29}$$

$\log(\frac{\pi^*_{k-1}(\mathbf{y})}{\pi^*_k(\mathbf{y})})$ can be simplified. We start from the definition provided in Equation 3 and ignored the policy update error $\pi_{\theta_{k-1}}(\mathbf{y}|\mathbf{x}) = \pi^*_{\theta_{k-1}}(\mathbf{y}|\mathbf{x})$ and $\tilde{\pi}^*_{\theta_k}(\mathbf{y}|\mathbf{x}) = \pi^*_{\theta_k}(\mathbf{y}|\mathbf{x})$:

$$\pi^*_{\theta_k}(\mathbf{y}|\mathbf{x}) = \frac{1}{Z_k(\mathbf{x})}\pi^*_{\theta_{k-1}}(\mathbf{y}|\mathbf{x})\exp\left(\frac{r_k}{\beta}\cdot\mathbb{I}(\mathbf{y}=\mathbf{y}_k)\right) \tag{30}$$

$$\frac{\pi^*_{\theta_k}(\mathbf{y})}{\pi^*_{\theta_{k-1}}(\mathbf{y})} = \frac{1}{Z_k(\mathbf{x})}\exp\left(\frac{r_k}{\beta}\cdot\mathbb{I}(\mathbf{y}=\mathbf{y}_k)\right) \tag{31}$$

$$\frac{\pi^*_{\theta_{k-1}}(\mathbf{y})}{\pi^*_{\theta_k}(\mathbf{y})} = \frac{Z_k(\mathbf{x})}{\exp\left(\frac{r_k}{\beta}\cdot\mathbb{I}(\mathbf{y}=\mathbf{y}_k)\right)} \tag{32}$$

$$\frac{\pi^*_{\theta_{k-1}}(\mathbf{y})}{\pi^*_{\theta_k}(\mathbf{y})} = Z_k(\mathbf{x})\exp\left(-\frac{r_k}{\beta}\cdot\mathbb{I}(\mathbf{y}=\mathbf{y}_k)\right) \tag{33}$$

Now, we take the natural logarithm of both sides of Equation 33:

$$\log\left(\frac{\pi^*_{\theta_{k-1}}(\mathbf{y})}{\pi^*_{\theta_k}(\mathbf{y})}\right) = \log\left(Z_k(\mathbf{x})\exp\left(-\frac{r_k}{\beta}\cdot\mathbb{I}(\mathbf{y}=\mathbf{y}_k)\right)\right) = \log(Z_k(\mathbf{x})) - \frac{r_k}{\beta}\mathbb{I}(\mathbf{y}=\mathbf{y}_k) \tag{34}$$

Substituting Equation 34 in:

$$D_{\mathrm{KL}}(\pi^*_{\mathrm{user}}\|\pi^*_{\theta_k}) - D_{\mathrm{KL}}(\pi^*_{\mathrm{user}}\|\pi^*_{\theta_{k-1}}) \tag{35}$$

$$= \sum_{\mathbf{y}}\pi^*_{\mathrm{user}}(\mathbf{y})\left[\log(Z_k(\mathbf{x})) - \frac{r_k}{\beta}\mathbb{I}(\mathbf{y}=\mathbf{y}_k)\right] \tag{36}$$

$$= \sum_{\mathbf{y}}\pi^*_{\mathrm{user}}(\mathbf{y})\log(Z_k(\mathbf{x})) - \sum_{\mathbf{y}}\pi^*_{\mathrm{user}}(\mathbf{y})\frac{r_k}{\beta}\mathbb{I}(\mathbf{y}=\mathbf{y}_k) \tag{37}$$

$$= \log(Z_k(\mathbf{x}))\left(\sum_{\mathbf{y}}\pi^*_{\mathrm{user}}(\mathbf{y})\right) - \frac{r_k}{\beta}\left(\sum_{\mathbf{y}}\pi^*_{\mathrm{user}}(\mathbf{y})\mathbb{I}(\mathbf{y}=\mathbf{y}_k)\right) \tag{38}$$

$$= \log(Z_k(\mathbf{x}))\cdot 1 - \frac{r_k}{\beta}\pi^*_{\mathrm{user}}(\mathbf{y}_k|\mathbf{x}) \tag{39}$$

$$= \log(Z_k(\mathbf{x})) - \frac{r_k}{\beta}\pi^*_{\mathrm{user}}(\mathbf{y}_k|\mathbf{x}) \tag{40}$$

$$\tag{41}$$

Given that the normalization constant $Z_k(\mathbf{x}) \leq 1$, it follows that $\log(Z_k(\mathbf{x})) \leq 0$. Furthermore, as the sample $\mathbf{y}_k$ is drawn from the user's target distribution $\pi^*_{\mathrm{user}}$, the reward is $r_k = 1$. Applying these conditions to Equation 41, we obtain the final inequality:

$$D_{\mathrm{KL}}(\pi^*_{\mathrm{user}}\|\pi^*_{\theta_k}) - D_{\mathrm{KL}}(\pi^*_{\mathrm{user}}\|\pi^*_{\theta_{k-1}}) \tag{42}$$

$$\leq 0 - \frac{1}{\beta}\pi^*_{\mathrm{user}}(\mathbf{y}_k|\mathbf{x})$$

$$= -\frac{1}{\beta}\pi^*_{\mathrm{user}}(\mathbf{y}_k|\mathbf{x}). \tag{43}$$

Since $\pi^*_{\mathrm{user}}(\mathbf{y}_k|\mathbf{x}) \geq 0$ and $\beta > 0$, the one-step change in KL divergence is less than or equal to zero. This completes the proof.

$\square$

## C.4 Proof of Theorem 3

*Proof of Theorem 3.* We want to bound the final estimation error after $K$ turns, $D_{\text{KL}}(\pi^*_{\text{user}}\|\tilde{\pi}^*_{\theta_K})$. We can express this final error as the initial error at turn 0 plus the sum of all one-step changes in error from turn 1 to $K$:

$$D_{\text{KL}}(\pi^*_{\text{user}}\|\tilde{\pi}^*_{\theta_K}) = D_{\text{KL}}(\pi^*_{\text{user}}\|\pi_{\theta_0}) + \sum_{k=1}^{K}\left(D_{\text{KL}}(\pi^*_{\text{user}}\|\tilde{\pi}^*_{\theta_k}) - D_{\text{KL}}(\pi^*_{\text{user}}\|\tilde{\pi}^*_{\theta_{k-1}})\right) \quad (44)$$

where we define $\tilde{\pi}^*_{\theta_0} = \pi_{\theta_0}$ as the initial policy.

From Theorem 2, we have an upper bound for each one-step change in error:

$$D_{\text{KL}}(\pi^*_{\text{user}}\|\tilde{\pi}^*_{\theta_k}) - D_{\text{KL}}(\pi^*_{\text{user}}\|\tilde{\pi}^*_{\theta_{k-1}}) \leq -\frac{1}{\beta}\pi^*_{\text{user}}(\mathbf{y}_k|\mathbf{x}) \quad (45)$$

We can apply this inequality to the summation term. By summing the upper bounds for each step from $k = 1$ to $K$, we get an upper bound for the total change:

$$\sum_{k=1}^{K}\left(D_{\text{KL}}(\pi^*_{\text{user}}\|\tilde{\pi}^*_{\theta_k}) - D_{\text{KL}}(\pi^*_{\text{user}}\|\tilde{\pi}^*_{\theta_{k-1}})\right) \leq \sum_{k=1}^{K}\left(-\frac{1}{\beta}\pi^*_{\text{user}}(\mathbf{y}_k|\mathbf{x})\right) \quad (46)$$

Substituting this bounded sum back into our expression for the final error, we arrive at the desired result:

$$D_{\text{KL}}(\pi^*_{\text{user}}\|\tilde{\pi}^*_{\theta_K}) \leq D_{\text{KL}}(\pi^*_{\text{user}}\|\pi_{\theta_0}) - \frac{1}{\beta}\sum_{k=1}^{K}\pi^*_{\text{user}}(\mathbf{y}_k|\mathbf{x}) \quad (47)$$

This completes the proof. $\square$

## C.5 Proof of Theorem 4

**Assumption 1** (Lipschitz-Smooth Log-Policy). *We assume the log-policy function $\log\pi_\theta$ is Lipschitz-smooth with constant $L$. This implies that the KL divergence between policies generated by two different parameter sets is bounded:*

$$D_{KL}(\pi_\theta\|\pi_{\theta'}) \leq \frac{L}{2}\|\theta - \theta'\|_2^2$$

*Proof.* Our goal is to bound the final error after $K$ turns, $D_{\text{KL}}(\pi^*_{\text{user}}\|\pi_{\theta_K})$. We begin by expressing this final error as the initial error plus the sum of all one-step changes:

$$D_{\text{KL}}(\pi^*_{\text{user}}\|\pi_{\theta_K}) = D_{\text{KL}}(\pi^*_{\text{user}}\|\pi_{\theta_0}) + \sum_{k=1}^{K}\left(D_{\text{KL}}(\pi^*_{\text{user}}\|\pi_{\theta_k}) - D_{\text{KL}}(\pi^*_{\text{user}}\|\pi_{\theta_{k-1}})\right)$$

The one-step change at turn $k$ can be decomposed by introducing our practical target policy, $\tilde{\pi}^*_{\theta_k}$, as an intermediate term:

$$D_{\text{KL}}(\pi^*_{\text{user}}\|\pi_{\theta_k}) - D_{\text{KL}}(\pi^*_{\text{user}}\|\pi_{\theta_{k-1}}) = \underbrace{D_{\text{KL}}(\pi^*_{\text{user}}\|\tilde{\pi}^*_{\theta_k}) - D_{\text{KL}}(\pi^*_{\text{user}}\|\pi_{\theta_{k-1}})}_{\text{Term A: Improvement from feedback}}$$
$$+ \underbrace{D_{\text{KL}}(\pi^*_{\text{user}}\|\pi_{\theta_k}) - D_{\text{KL}}(\pi^*_{\text{user}}\|\tilde{\pi}^*_{\theta_k})}_{\text{Term B: Error from inexact update}}$$

We now bound these two terms separately.

**Bounding Term A (Improvement):** From Theorem 2, we have a direct upper bound for the first term, which represents the guaranteed error reduction from applying the user feedback to form the new target:

$$D_{\text{KL}}(\pi^*_{\text{user}}\|\tilde{\pi}^*_{\theta_k}) - D_{\text{KL}}(\pi^*_{\text{user}}\|\pi_{\theta_{k-1}}) \leq -\frac{1}{\beta}\pi^*_{\text{user}}(\mathbf{y}_k|\mathbf{x})$$

**Bounding Term B (Approximation Error):** The second term represents the error introduced because our updated policy $\pi_{\theta_k}$ is not exactly equal to the practical target $\tilde{\pi}_{\theta_k}^*$ due to the linearization in our parameter update step. We can bound this term using the smoothness assumption. A key property of KL divergence is that $D_{\text{KL}}(P\|Q) - D_{\text{KL}}(P\|R)$ is related to $D_{\text{KL}}(R\|Q)$. Specifically, the error introduced by our inexact update $\pi_{\theta_k} \approx \tilde{\pi}_{\theta_k}^*$ can be bounded by the KL divergence between them, which in turn is bounded by the squared norm of the parameter update step under Assumption 1:

$$D_{\text{KL}}(\pi_{\text{user}}^*\|\pi_{\theta_k}) - D_{\text{KL}}(\pi_{\text{user}}^*\|\tilde{\pi}_{\theta_k}^*) \leq D_{\text{KL}}(\tilde{\pi}_{\theta_k}^*\|\pi_{\theta_k}) \leq \frac{L}{2}\|\Delta\theta_k\|_2^2$$

This is a standard result from analyzing the convergence of mirror descent, where our update is an instance.

**Combining the Bounds:** We can now sum the bounds for Term A and Term B over all $K$ turns:

$$\sum_{k=1}^{K}\left(D_{\text{KL}}(\pi_{\text{user}}^*\|\pi_{\theta_k}) - D_{\text{KL}}(\pi_{\text{user}}^*\|\pi_{\theta_{k-1}})\right)$$

$$\leq \sum_{k=1}^{K}\left(-\frac{1}{\beta}\pi_{\text{user}}^*(\mathbf{y}_k|\mathbf{x}) + \frac{L}{2}\|\Delta\theta_k\|_2^2\right)$$

$$= -\frac{1}{\beta}\sum_{k=1}^{K}\pi_{\text{user}}^*(\mathbf{y}_k|\mathbf{x}) + \frac{L}{2}\sum_{k=1}^{K}\|\Delta\theta_k\|_2^2$$

Substituting this summed bound back into our initial expression for the final error, we arrive at the unified convergence bound:

$$D_{\text{KL}}(\pi_{\text{user}}^*\|\pi_{\theta_K}) \leq D_{\text{KL}}(\pi_{\text{user}}^*\|\pi_{\theta_0}) - \frac{1}{\beta}\sum_{k=1}^{K}\pi_{\text{user}}^*(\mathbf{y}_k|\mathbf{x}) + \frac{L}{2}\sum_{k=1}^{K}\|\Delta\theta_k\|_2^2$$

This completes the proof. $\square$

# D EXPERIMENTAL SETTING

We conduct a comprehensive evaluation of ROSA across a diverse set of tasks and models to validate its generalizability, effectiveness, and efficiency.

## D.1 DATASETS.

To demonstrate the broad applicability of ROSA, we select challenging benchmarks spanning four distinct problem-solving domains. A summary of these datasets is provided in Table 4, followed by detailed descriptions.

Table 4: Overview of the datasets used for evaluation. "N/A" indicates that the dataset is primarily for evaluation and does not have a standard, predefined training set.

| Domain | Dataset Name | Training Size | Test Size |
|---|---|---|---|
| | MATH | 7,500 | 5,000 |
| Mathematical Reasoning | AIME25 | N/A | 30 |
| | MATH-500 | N/A | 500 |
| | GPQA-diamond | N/A | 198 |
| General Reasoning | MMLU-Redux | N/A | 3,000 |
| | SuperGPQA | 26,500 | N/A |
| Code Generation | HumanEval | N/A | 164 |
| Multilingual Reasoning | MCLM | N/A | 156 |

**Mathematical Reasoning.** This domain focuses on complex, multi-step mathematical problem-solving. We use three standard benchmarks. MATH (Hendrycks et al., 2021b) is a dataset of 12,500 challenging competition mathematics problems from high school level, covering topics like algebra, geometry, and calculus. AIME25 (AIME, 2025) is a curated set of 25 highly difficult problems from the American Invitational Mathematics Examination (AIME), designed to test advanced reasoning capabilities. MATH-500 (Lightman et al., 2023) is a well-known evaluation subset of the MATH test set, consisting of 500 problems often used for efficient model assessment.

**General Reasoning.** To evaluate reasoning on a broad range of topics, we use three expert-level question-answering datasets. GPQA-diamond (Rein et al., 2024) is a challenging set of graduate-level, Google-proof questions written by domain experts, where the "diamond" subset represents the highest-quality questions. MMLU-Redux (Hendrycks et al., 2021a) is a revised and cleaned version of the Massive Multitask Language Understanding benchmark, which covers 57 diverse subjects from elementary mathematics to US history and law. SuperGPQA (Team et al., 2025) significantly expands upon GPQA, containing nearly 5,000 expert-validated questions across 285 graduate-level disciplines.

**Code Generation.** We test the ability of models to generate functionally correct code from natural language descriptions using HumanEval (Chen et al., 2021). This dataset consists of 164 hand-written programming problems with function signatures, docstrings, and unit tests to verify the correctness of the generated code.

**Multilingual Reasoning.** To assess reasoning capabilities across different languages, we use the MCLM (Son et al., 2025) benchmark. This dataset was created by translating challenging English reasoning benchmarks into multiple languages. Our evaluation focuses on its subsets, including multilingual versions of IMO, AIME, and MATH problems (M-IMO, MT-AIME24, and MT-MATH100).

**Dataset Usage in Experiments.** Our primary evaluation of effectiveness of ROSA is conducted on official, held-out test sets to simulate real-world performance. For experiments where a dedicated test set is not available, or for ablation studies, we utilize the corresponding training or development sets for analysis. This ensures a comprehensive assessment of ROSA capabilities across different conditions while maintaining a clear distinction between final evaluation and component analysis. Specifically, we only sample part of the data from the SuperGPQA training set for testing, and the rest of the data sets are tested on the test set.

## D.2 MODELS

Our evaluation includes a variety of recent open-source LLMs to ensure our findings are not model-specific. These models are selected to cover a range of sizes and specializations, as summarized in Table 5 and detailed below. To mitigate potential data contamination issues with the Qwen2.5 series on certain benchmarks, we also conduct validation experiments on the more recent Qwen3 and DeepSeek-R1 models. All models used are instruction-tuned variants designed for chat and instruction-following tasks.

Table 5: Overview of the language models used in our experiments, categorized by scale and specialization.

| Category | Model Name | Parameters | Variant |
|---|---|---|---|
| Small-Scale Models | Qwen2.5-0.5B-Instruct | 0.5B | Instruct |
|  | Qwen3-0.6B | 0.6B | Base |
| Large-Scale Models | Qwen2.5-7B-Instruct | 7B | Instruct |
|  | Qwen3-8B | 8B | Base |
| Reasoning-Focused | DeepSeek-R1-Distill-Llama-8B | 8B | Reasoning-Tuned |
|  | DeepSeek-R1-Distill-Qwen-7B | 7B | Reasoning-Tuned |

**Small-Scale Models.** To assess the performance of ROSA on more compact models, we selected two from the Qwen family, known for their strong general-purpose capabilities. `Qwen2.5-0.5B-Instruct` (Qwen et al., 2025) is a 0.5 billion parameter model from the Qwen2.5 series, optimized for instruction following. `Qwen3-0.6B` (Yang et al., 2025a) is a 0.6 billion parameter model from the newer Qwen3 generation, featuring architectural improvements.

**Large-Scale Models.** We evaluate on larger, more capable base models to test the scalability of our approach. These include `Qwen2.5-7B-Instruct` (Qwen et al., 2025), a widely-used 7 billion parameter instruction-tuned model, and `Qwen3-8B` (Yang et al., 2025a), its 8 billion parameter successor from the Qwen3 series.

**Reasoning-Focused Models.** To specifically test performance on complex reasoning, we use models from the DeepSeek-R1 series, which are explicitly optimized for reasoning capabilities through reinforcement learning (DeepSeek-AI et al., 2025). The models we use are distilled versions of a larger, proprietary model. `DeepSeek-R1-Distill-Llama-8B` is an 8 billion parameter model that uses a Llama-based architecture. `DeepSeek-R1-Distill-Qwen-7B` is a 7 billion parameter variant that is instead based on the Qwen architecture, allowing for a more controlled comparison with the general-purpose Qwen models.

### D.3 EVALUATION METRICS

We assess ROSA based on two primary aspects: performance and efficiency.

**Performance Metrics.** To measure problem-solving success, we define two key metrics. *Accuracy* is the final proportion of unique problems solved correctly within a total of $K$ conversational turns. Let $\mathcal{P}$ be the set of all problems, and let $S_i \in \{0, 1\}$ be an indicator variable where $S_i = 1$ if problem $i$ is solved at any turn up to $K$. The accuracy is given by:

$$\text{Accuracy} = \frac{\sum_{i \in \mathcal{P}} S_i}{|\mathcal{P}|} \tag{48}$$

*Correction Uplift* measures the ability of model to recover from initial failures. It is the percentage of problems that were answered incorrectly in the first turn but were successfully corrected in a subsequent turn. Let $\mathcal{P}_{\text{fail}} \subset \mathcal{P}$ be the subset of problems that the model failed to solve in the first turn. The Correction Uplift is:

$$\text{Correction Uplift} = \frac{\sum_{i \in \mathcal{P}_{\text{fail}}} S_i}{|\mathcal{P}_{\text{fail}}|} \times 100\% \tag{49}$$

**Efficiency Metrics.** To evaluate the computational overhead of our method, we measure two metrics. *Latency* is the average wall-clock time required for a single generation and update cycle. *Peak GPU Memory* is the maximum GPU memory consumed during this cycle. These metrics are crucial for assessing the practical feasibility of deploying ROSA in real-world interactive systems.

### D.4 REWARD MODELS

To simulate different real-world feedback scenarios, we employ two types of reward models.

**Rule-Based Reward Model.** This model simulates scenarios with definitive, high-level judgments by providing a sparse feedback signal of $\{-1, +1\}$. It programmatically extracts the final answer from a model's response, typically from a `\boxed{}` environment, and compares it to the ground-truth solution. A reward of $+1.0$ is assigned for a correct answer, and $-1.0$ otherwise. This mimics situations where feedback is based solely on the final outcome. The core logic implementation is shown in the following table.

**Core logic for the rule-based reward model.**

```python
class MathVerifyRewardModel:
    def __init__(self, ground_truth_answer: str):
        self.ground_truth_answer = ground_truth_answer

    def get_reward(self, response_text: str) -> float:

        return 1.0 if compute_score(response_text,
        self.ground_truth_answer) == 1.0 else -1.0

def compute_score(solution_str, ground_truth) -> float:
    retval = 0.0
    try:
        string_in_last_boxed =
        last_boxed_only_string(solution_str)
        if string_in_last_boxed is not None:
            answer = remove_boxed(string_in_last_boxed)
            if is_equiv(answer, ground_truth):
                retval = 1.0
    except Exception:
        pass
    return retval
```

**Model-Based Reward Model.** This model simulates more nuanced, fine-grained human feedback by providing a dense, continuous reward score in the range $[-1.0, +1.0]$. We use a powerful, proprietary large language model, `Qwen/Qwen3-30B-A3B-Instruct-2507`, as the reward judge. The model is deployed using the VLLM inference engine for efficient scoring. It evaluates the generated response based on correctness, reasoning, and style by comparing it against the problem statement and the ideal solution. The prompt used to elicit the score is shown in the following table.

**The prompt template for the model-based reward system.**

```
A student AI was asked the following problem: {problem}.
The student AI gave the following answer: {generated_text}.
The ideal correct solution and answer is {solution}.
Please grade strictly but fairly.
Compare the student's answer to the ideal answer.
Evaluate the student's answer based on correctness, reasoning,
and style.
Note: Based on your evaluation, please provide a floating point
score from -1.0 (completely wrong) to 1.0 (perfect).
The score should be placed at the end of your answer in the
format: SCORE: [score].
```

### D.5 PARAMETER UPDATE MECHANISMS

To implement the policy update $\Delta\theta$ computed in Section 3.3, we introduce two distinct, lightweight update mechanisms. These methods are designed to be computationally efficient, allowing for real-time policy adaptation during the inference phase without significant overhead.

**1. LM Head Update via LoRA.** The first mode targets the final layer of the model, the language modeling (LM) head. The LM head is typically a linear layer (an MLP matrix) that projects the final hidden state representation of the model into the vocabulary space to produce logits. We augment

this layer by adding a Low-Rank Adaptation (LoRA) (Hu et al., 2022) matrix. Specifically, a low-rank decomposition, represented by two matrices $\mathbf{A} \in \mathbb{R}^{d \times r}$ and $\mathbf{B} \in \mathbb{R}^{r \times V}$ (where $d$ is the hidden size, $V$ is the vocabulary size, and $r \ll d, V$ is the rank), is added to the original LM head weight matrix. During our online update process, only the parameters of these small LoRA matrices $\mathbf{A}$ and $\mathbf{B}$ are modified. The parameter update $\Delta\theta$ calculated by the CG solver is applied directly to the flattened weights of $\mathbf{A}$ and $\mathbf{B}$. This approach confines the policy optimization to a very small subset of the total model parameters, preserving the model's foundational knowledge while enabling rapid and efficient adaptation of its final output probabilities. The specific LoRA configuration is shown in Table 6.

Table 6: LoRA Hyperparameter Configuration.

| Hyperparameter | Value |
|---|---|
| target_modules | lm_head |
| Rank | 1 |
| lora_alpha | 8 |
| lora_dropout | 0.1 |

**2. Hidden State Modification.** The second mode operates not on the model's weights, but directly on its activations (Hu et al., 2025). Instead of modifying a layer, we intercept the final hidden state $\mathbf{H} \in \mathbb{R}^{1 \times d}$ just before it is passed to the LM head. We then compute an update vector $\Delta\mathbf{H} \in \mathbb{R}^{1 \times d}$ (which in this context represents our $\Delta\theta$) and add it directly to the hidden state to produce a modified activation:

$$\mathbf{H}_{\text{new}} = \mathbf{H} + \Delta\mathbf{H} \tag{50}$$

This new hidden state, $\mathbf{H}_{\text{new}}$, is then passed to the original, unmodified LM head to generate the final logits. This method is implemented using model hooking techniques, which allow us to register a forward hook on the LM head layer. The hook intercepts the input ($\mathbf{H}$), applies the additive modification, and returns the transformed tensor as the new input for the layer's forward pass. This approach completely avoids any updates to the persistent model weights and instead performs a transient, state-dependent policy correction on the activation flow.

# E  MORE RESULT

## E.1  ADDITIONAL EMPIRICAL RESULTS

This section presents supplementary empirical results to further validate our findings. First, Table 8 reports the performance of all models on three benchmarks—AIME25, GPQA-diamond, and M-IMO—which were omitted from the main text due to space constraints. Second, to provide a more complete picture of model performance, Table 7 details the *Accuracy* and *Correction Uplift* for the DeepSeek-R1-Distill-Qwen-7B model on both mathematical reasoning and code generation datasets. Across these additional results, a clear and consistent trend emerges: reinforcing the conclusions from our main analysis, our proposed method, ROSA, significantly enhances both overall task performance and the capacity of model for self-correction.

Table 7: Performance of the **DeepSeek-R1-Distill-Qwen-8B** model on mathematical reasoning and code generation datasets. The values in red indicate the absolute improvement of ROSA over the baseline.

| | MATH | | AIME25 | | MATH-500 | | HumanEval | |
|---|---|---|---|---|---|---|---|---|
| Method | Final Acc. ↑ | Correction Uplift ↑ | Final Acc. ↑ | Correction Uplift ↑ | Final Acc. ↑ | Correction Uplift ↑ | Final Acc. ↑ | Correction Uplift ↑ |
| Baseline | 7.60 | 3.14 | 10.00 | 3.57 | 7.40 | 6.09 | 45.12 | 17.05 |
| ROSA | 9.80 (+2.20) | 5.65 (+2.51) | 16.67 (+6.67) | 16.67 (+13.10) | 22.20 (+14.80) | 18.62 (+12.53) | 51.22 (+6.10) | 33.75 (+16.70) |

## E.2  COMPARISON WITH MULTI-TURN TRAINING METHODS

While our main analysis focuses on test-time adaptation, it is instructive to compare ROSA with traditional training-based methods for multi-turn dialogue. In this section, we benchmark the per-

Table 8: Supplementary performance results on additional benchmarks, reporting accuracy (%). The values in red indicate the absolute improvement of ROSA variants over the baseline.

| Model | Method | Mathematical Reasoning | General Reasoning | Multilingual Reasoning |
|---|---|---|---|---|
| | | AIME25 | GPQA-diamond | M-IMO |
| Qwen2.5-0.5B -Instruct | Baseline | 3.33 | 3.54 | 1.99 |
| | ROSA (+LM + R) | **6.67** (+3.34) | 7.07 (+3.53) | 2.09 (+0.10) |
| | ROSA (+HS + R) | **6.67** (+3.34) | 8.53 (+4.99) | 3.20 (+1.21) |
| | ROSA (+LM + M) | **6.67** (+3.34) | 10.27 (+6.73) | 4.71 (+2.72) |
| Qwen3-0.6B | Baseline | 10.00 | 12.20 | 5.20 |
| | ROSA (+LM + R) | **16.67** (+6.67) | 9.09 (+-3.11) | 5.30 (+0.10) |
| | ROSA (+HS + R) | 10.00 (+0.00) | 10.54 (+-1.66) | 5.30 (+0.10) |
| | ROSA (+LM + M) | 10.00 (+0.00) | 13.16 (+0.96) | 6.60 (+1.40) |
| Qwen2.5-7B -Instruct | Baseline | 10.00 | 26.14 | 10.53 |
| | ROSA (+LM + R) | **23.33** (+13.33) | 42.24 (+16.10) | 17.57 (+7.04) |
| | ROSA (+HS + R) | 20.00 (+10.00) | 43.16 (+17.02) | 18.36 (+7.83) |
| | ROSA (+LM + M) | 20.00 (+10.00) | 45.83 (+19.69) | 21.21 (+10.68) |
| Qwen3-8B | Baseline | 16.67 | 41.16 | 20.37 |
| | ROSA (+LM + R) | 30.00 (+13.33) | 69.11 (+27.95) | 33.17 (+12.80) |
| | ROSA (+HS + R) | 33.33 (+16.66) | 70.27 (+29.11) | 37.62 (+17.25) |
| | ROSA (+LM + M) | **36.67** (+20.00) | 75.18 (+34.02) | 39.16 (+18.79) |
| DeepSeek-R1 -Distill-Llama-8B | Baseline | 3.33 | 19.03 | 4.36 |
| | ROSA (+LM + R) | **16.67** (+13.34) | 21.14 (+2.11) | 6.32 (+1.96) |
| | ROSA (+HS + R) | **16.67** (+13.34) | 22.23 (+3.20) | 5.17 (+0.81) |
| | ROSA (+LM + M) | **16.67** (+13.34) | 25.36 (+6.33) | 6.36 (+2.00) |

formance of ROSA against two such paradigms on the MATH dataset: Supervised Fine-Tuning (SFT) and Reinforcement Learning (RL).

For the SFT baseline, we first generated a multi-turn dialogue dataset using DeepSeek-R1 on the MATH training set, and then fine-tuned the base model on this newly created data. For the RL baseline (Sheng et al., 2025), we employed a Group Preference Optimization (GRPO) scheme tailored for multi-turn dialogue, similar to the approach described in our related work (Appendix B).

The results, presented in Table 9, report both *Accuracy* and *Correction Uplift*. The key finding is that ROSA, a purely test-time method, achieves performance that is comparable or even superior to these training-based approaches. This highlights a significant advantage of our method: it obviates the need for expensive data collection and resource-intensive model training, offering a more efficient and flexible solution for enhancing multi-turn capabilities.

Table 9: Comparison of ROSA with training-based methods on the **MATH** dataset for the **Qwen3-8B** model. Our test-time method achieves performance comparable to full Reinforcement Learning (RL) training and surpasses Supervised Fine-Tuning (SFT), without requiring data collection or model training.

| Method | Final Acc. ↑ | Correction Uplift ↑ |
|---|---|---|
| Baseline | 55.80 | 23.00 |
| SFT Training | 63.80 | 39.24 |
| RL Training | **66.20** | **40.45** |
| ROSA | 65.80 | 40.42 |

### E.3 EFFICIENCY ANALYSIS

In this section, we analyze the computational overhead of ROSA in Table 10. ROSA introduces an explicit parameter update step, which incurs additional time and memory costs. As shown in the table 10, the *Avg. Update Time* makes the total processing time per turn roughly double that of the baseline's inference-only time. However, this update process is designed to be executed asynchronously. In a real-world application, the update can be performed in the background while the user is interpreting the response of model and formulating their next prompt. This parallel

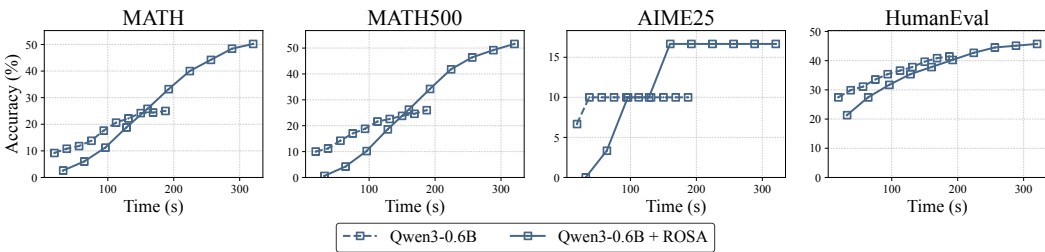

Figure 6: Time-to-Accuracy comparison for the `Qwen3-0.6B` model with (ROSA) and without (Baseline) our method on the MATH, MATH500, AIME25, and HumanEval datasets. The x-axis represents the cumulative wall-clock time in seconds. Our method ROSA consistently has larger slopes, highlighting its significant advantage in time efficiency.

processing makes the update latency largely **imperceptible to the user**, enabling a seamless and responsive interactive experience.

We plotted time and accuracy as a line graph, as shown in Figure 6, clearly demonstrating the time efficiency of our method. The graph plots accuracy as a function of cumulative time. A consistent trend can be observed across all four benchmarks: the curve for our method, ROSA (solid line), has a significantly steeper slope than the baseline (dashed line). This indicates a faster rate of improvement in accuracy per second, validating ROSA's effectiveness in conversational error correction. Notably, even on datasets where ROSA initially had lower accuracy (such as MATH and MATH500), its superior error correction efficiency enabled it to quickly surpass the baseline. Ultimately, our method not only achieved a higher final accuracy ceiling, but also achieved this in a shorter time, highlighting its practical advantages in real-world interaction scenarios.

Regarding memory, the *Update Peak* shows only a modest increase over the *Inference Peak*. This demonstrates that ROSA can perform its online updates **without a prohibitive increase in GPU memory requirements**, confirming its practicality for deployment on existing hardware.

Table 10: Efficiency analysis of ROSA. We report the averaged inference latency and peak GPU memory per turn. The "Update" columns show the additional overhead introduced by ROSA.

| Model | Method | Averaged Latency (s) | | Peak GPU Memory (GB) | |
|---|---|---|---|---|---|
| | | Inference ↓ | Update ↓ | Inference ↓ | Update ↓ |
| Qwen2.5-0.5B -Instruct | Baseline | 8.42 | 0 | 11.85 | 0 |
| | ROSA | 7.97 | 7.58 | 11.68 | 18.60 |
| Qwen3-0.6B | Baseline | 18.71 | 0 | 15.61 | 0 |
| | ROSA | 18.85 | 13.16 | 16.24 | 23.65 |
| Qwen2.5-7B -Instruct | Baseline | 19.40 | 0 | 42.01 | 0 |
| | ROSA | 19.72 | 23.28 | 43.41 | 48.95 |
| Qwen3-8B | Baseline | 30.64 | 0 | 51.90 | 0 |
| | ROSA | 30.70 | 28.39 | 51.90 | 58.63 |
| DeepSeek-R1 -Distill-Llama-8B | Baseline | 26.40 | 0 | 48.16 | 0 |
| | ROSA | 26.70 | 26.37 | 48.73 | 54.8 |
| DeepSeek-R1 -Distill-Qwen-7B | Baseline | 23.53 | 0 | 44.27 | 0 |
| | ROSA | 23.56 | 25.87 | 45.52 | 51.61 |

### E.4 ABLATION STUDIES

#### E.4.1 THE IMPORTANCE OF THE OPTIMIZATION STRATEGY

To isolate the contribution of our proposed optimization method, we conduct an ablation study comparing the full ROSA framework against a more direct reinforcement learning approach. This baseline, which we term *RL*, directly optimizes the standard RLHF objective function in (1)). To

Table 11: Comparison of Accuracy (%) on mathematical reasoning datasets with RL and ROSA.

| Model | Method | MATH | MATH-500 | AIME25 | HumanEval |
|---|---|---|---|---|---|
| Qwen3-0.6B | RL | 26.20 | 28.80 | 10.00 | 42.68 |
| | ROSA | **50.20** (+24.00) | **51.60** (+22.80) | **16.67** (+6.67) | **45.73** (+3.05) |
| Qwen3-8B | RL | 59.60 | 63.60 | 16.67 | 79.27 |
| | ROSA | **65.80** (+6.20) | **72.80** (+9.20) | **30.00** (+13.33) | **81.71** (+2.44) |
| DeepSeek-R1-Distill -Llama-8B | RL | 6.20 | 8.40 | 10.00 | 28.05 |
| | ROSA | **7.80** (+1.60) | **18.40** (+10.00) | **16.67** (+6.67) | **39.02** (+10.97) |

Table 12: Comparison of Correction Uplift (%) on mathematical reasoning datasets with RL and ROSA.

| Model | Method | MATH | MATH-500 | AIME25 | HumanEval |
|---|---|---|---|---|---|
| Qwen3-0.6B | RL | 18.54 | 20.00 | 6.90 | 22.31 |
| | ROSA | **48.87** (+30.33) | **51.31** (+31.31) | **16.67** (+9.77) | **31.01** (+8.70) |
| Qwen3-8B | RL | 29.37 | 31.58 | 10.71 | 50.00 |
| | ROSA | **40.42** (+11.05) | **52.94** (+21.36) | **27.59** (+16.88) | **62.50** (+12.50) |
| DeepSeek-R1-Distill -Llama-8B | RL | 4.87 | 7.29 | 6.90 | 16.31 |
| | ROSA | **6.30** (+1.43) | **17.41** (+10.12) | **13.79** (+6.89) | **31.97** (+15.66) |

simulate the online, multi-turn interaction setting in a comparable manner to ROSA, we estimate the gradient of $J(\pi_\theta)$ using only a single response $\mathbf{y}$ sampled from the policy $\pi_\theta$ for each prompt $\mathbf{x}$, and then update the model's parameters using this gradient. This approach contrasts with our full ROSA framework, which first computes a stable target policy $\pi^*$ and then solves for the parameter update $\Delta\theta$.

The results of this comparison are presented in Figure 4. The analysis leads to two clear observations. First, the direct RL optimization (dotted lines) yields only marginal improvements over the baseline models (solid lines) across all three datasets. The proximity of the solid and dotted lines indicates that a naive policy gradient update with a single sample provides a noisy and inefficient learning signal, resulting in minimal performance gains. Second, in stark contrast, ROSA (dashed lines) consistently and significantly outperforms both the baseline and the RL-enhanced version. The steeper slopes of the dashed lines demonstrate that ROSA not only achieves a higher absolute accuracy but also accelerates the error correction process over the conversation turns. For example, on the MATH dataset, the Qwen3-8B model enhanced with ROSA shows a much more rapid accuracy improvement compared to its *RL* counterpart.

The quantitative results of this comparison, presented in Table 11 and Table 12, demonstrate a clear and consistent advantage for ROSA. Table 11 reveals that ROSA achieves substantially higher final accuracy across all models and datasets. For instance, on the MATH dataset with the Qwen3-0.6B model, ROSA surpasses the RL baseline by a remarkable **+24.00%**. Furthermore, Table 12 highlights its superior self-correction capability. In the most significant case, ROSA boosts the Correction Uplift score by **+31.31%** on MATH-500 for the same model. The data consistently show that a direct RL update provides only marginal benefits, while our principled optimization strategy yields significant gains in both overall success and the ability to recover from errors.

This ablation study confirms that the superior performance of ROSA is not merely due to the introduction of an online reward signal. Rather, it is the principled optimization strategy—deriving a stable online target $\pi^*$ and then efficiently solving for the optimal parameter update $\Delta\theta$—that is crucial for achieving effective and efficient test-time adaptation.

### E.4.2 ABLATION STUDY ON THE INFLUENCE OF HYPERPARAMETER $\beta$

**Experimental Setup and the Role of $\beta$.** To investigate the sensitivity of our proposed method to its hyperparameters, we conduct an ablation study on the regularization coefficient $\beta$. We vary its value across a wide range of $[0.25, 1.75]$ to observe its impact on model performance. As defined in the standard RLHF objective, $\beta$ controls the trade-off between maximizing the reward and maintaining proximity to the reference policy. In the ROSA framework, its role is to modulate the intensity of the policy update based on the reward signal $r(\mathbf{x}, \mathbf{y})$, as formulated in our practical update target

in (3).A smaller $\beta$ amplifies the reward signal, leading to more aggressive updates, while a larger $\beta$ dampens it, resulting in more conservative updates.

**Analysis and Conclusions.** The results of our study are presented in Figure 7, which illustrates the cumulative accuracy over 10 conversational turns for each tested $\beta$ value. A key observation is that while the initial learning trajectories vary—with smaller $\beta$ values often yielding a steeper initial performance gain—all configurations converge to a similar final accuracy. This convergence can be attributed to the iterative nature of the multi-turn interaction. Although $\beta$ adjusts the magnitude of each corrective step, the consistent directional feedback provided by the reward signal ensures that the model is always guided towards an improved policy. Consequently, over a sufficient number of turns, even a series of conservative updates can accumulate to achieve the correct solution.

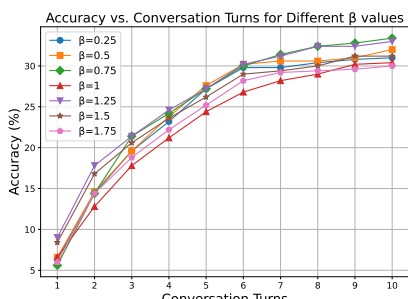

Figure 7: **Ablation study of the hyperparameter** $\beta$ **on the MATH dataset.** The figure illustrates the cumulative accuracy over 10 conversational turns for different values of $\beta$, ranging from 0.25 to 1.75.

From this analysis, we draw two key conclusions. First, for tasks with definitive solutions, such as mathematical reasoning, different search strategies—ranging from aggressive to conservative—are all highly likely to converge to the correct solution given adequate opportunities for self-correction. Second, this study underscores the **robustness of the ROSA framework**. The model's final performance demonstrates low sensitivity to the choice of $\beta$ across a wide operational range, indicating that ROSA can achieve stable and effective results without extensive hyperparameter tuning.

# F    CASE STUDY

## Qwen3-0.8B, MATH dataset

```
Question: Convert the point (0,3) in rectangular coordinates to polar coordinates.
Enter your answer in the...

True answer: \left( 3, \frac{\pi}{2} \right)

# --- Turn 1 ---
    Baseline answer: 'Okay, so I need to convert the point (0, 3) from rectangular
    coordinates to polar coordinates. Hmm, let me remember how to do this....
    So putting it all together, the polar coordinates should be (3, pi/2).
    Let me just'

    ROSA answer: 'Okay, so I need to convert the point (0, 3) from rectangular
    coordinates to polar coordinates. Hmm, let me remember how to do this....
    So putting it all together, the polar coordinates should be (3, pi/2).
    Let me just'

# --- Turn 2 ---

    Baseline answer: 'Okay, let me try to figure this out again. The original
    point is (0, 3). In rectangular coordinates....Therefore, after checking,
    I think the correct answer is (3, pi/2). However, the user initially
    thought it was wrong, but after rethinking, it's correct.'

    ROSA answer: 'Okay, so I got confused earlier. The original point is (0, 3)
    . If I use polar coordinates,....Alternatively, if someone thought that
    theta is measured as the angle between the x-axis and the'

# --- Turn 3 ---
    Baseline answer: 'Okay, so the user initially thought that converting
    (0,3) to polar coordinates woul....Is there any possibility that theta
    could be negative? No, because theta is always between 0 and 2pi,
    and since (0,3) is in the first quadrant'

    ROSA answer: 'Okay, so the user initially thought the answer was
    (3, pi/2) but the second time they said "Wrong answer....
    **Final Answer**
    The polar coordinates of the point (0, 3)
    are \boxed{(3, \frac{\pi}{2})}.'
```