# OpenReview forum: "Test-Time Policy Adaptation for Enhanced Multi-Turn Interactions with LLMs"
_ICLR.cc/2026/Conference — Submitted to ICLR 2026_

### Official Review · Reviewer_etZK · 2025-10-26

**Soundness:** 2
**Presentation:** 2
**Contribution:** 2
**Rating:** 2
**Confidence:** 3

**Summary:**

This paper identifies a key weakness in modern LLMs: their performance degrades in multi-turn interactions because they are typically trained on static, single-turn data and cannot adapt to real-time user feedback. To address this, the authors propose a new paradigm, "Test-Time Policy Adaptation for Multi-Turn Interactions" (T2PAM), where the model's policy is updated "in-conversation" based on user feedback. They also introduce ROSA (Optimum-Referenced One-Step Adaptation), a specific algorithm to implement T2PAM. ROSA uses user feedback (e.g., "wrong answer," mapped to a reward) to analytically compute an optimal target policy and then performs a single, efficient update step to steer the model's parameters (via linearized optimization) toward this target. The paper also provides theoretical analysis to guarantee convergence.

**Strengths:**

The primary strength of this paper is its motivation. The problem it addresses—that models are static at test time and fail to learn from immediate user feedback during an interaction—is highly relevant. The goal of enabling real-time policy adaptation to correct errors in a multi-turn setting is a valuable and important research direction.

**Weaknesses:**

Despite the strong motivation, this paper suffers from several major flaws in its execution, theoretical justification, and experimental evaluation.

- Presentation and Focus: The paper's structure feels imbalanced. A significant portion of the main text is dedicated to theoretical analysis (Section 4). However, the experimental results section (Section 5) is comparatively brief, and critical details about baselines and efficiency are relegated to the appendix. This suggests a desire for the theory to be the main contribution, but the theoretical claims themselves appear to have issues (see below). Important information, such as accurate descriptions of the baselines, should be in the main body for a clear comparison.

- Computational Cost: The user's concern about "computation time" is valid. The paper's own efficiency analysis (Appendix E.3, Table 10) shows that the "Avg. Update Time" is substantial, roughly equal to the "Inference" time. This means the method effectively doubles the computational cost of each turn. The claim that this is "imperceptible to the user" because it can be run asynchronously is a very strong assumption that may not hold in practice, where users expect immediate responses.

- Limited Problem Formulation: The "multi-turn" task is not representative of typical multi-turn interactions. The paper frames the task as correcting a final answer to a static problem. In contrast, many (if not most) multi-turn tasks involve a process that requires multiple steps to solve (e.g., planning, complex QA, collaborative writing). In these more realistic scenarios, the paper's assumption of receiving clear, accurate, and immediate feedback (e.g., r = -1) at each intermediate step is highly impractical, as the feedback itself would likely be noisy, delayed, or unavailable until the very end.

**Questions:**

- There appears to be a contradiction in the proof of Theorem 2. The T2PAM paradigm explicitly states that the policy update is only triggered by failure, i.e., negative feedback ($r_k = -1$). However, the proof of monotonic error reduction (Appendix C.3, lines 1068-1069) explicitly assumes a positive reward ($r_k = 1$) to complete the bound. Is this a typo or its my misunderstanding?

- How could this method be extended to scenarios where intermediate feedback is not accurately or immediately available?

---

> ### Author Response · Authors · 2025-11-16
> **Responses to Questions 1**
>
> We fully agree that the notation here can be confusing. This is not a fundamental contradiction, but rather a notational distinction that requires clarification. We will explicitly differentiate these in the final version.
>
> * **Clarification:** Your understanding is correct: the $T^2PAM$ paradigm **only triggers a parameter update upon receiving failure feedback (i.e., an observed reward $r_{k}=-1$)**.
>
> * **Logic of the Proof:** However, Theorem 2 and its proof (Appendix C.3) are analyzing the KL divergence $D_{\mathrm{KL}}(\pi_{user} \parallel \pi_{\theta_{k}})$—that is, the divergence between our model's adapted policy $\pi_{\theta_{k}}$ and the **optimal user policy** $\pi_{user}$.
>     * By the definition of KL divergence, $D_{\mathrm{KL}}(P \parallel Q) = \sum_{y} P(y) \log \frac{P(y)}{Q(y)}$, the expectation is computed over the "true" distribution $P$. In our analysis, this $P$ is the optimal user policy, $\pi_{user}$.
>     * Therefore, the $y_k$ in the proof's summation represents a sample drawn from $\pi_{user}(\cdot|x)$—that is, given a problem $x$, $y_k$ is a response sampled from the user's **ideal target distribution**.
>     * **The key point is:** $\pi_{user}$ is defined as the policy that "represents the true preferences and the ideal way to solve the task". By this very definition, any response $y_k$ sampled from $\pi_{user}(\cdot|x)$ is an *ideal, correct* response to $x$. Consequently, the reward $r_k$ associated with this sample *within the theoretical expectation of the KL divergence* must be $+1$. This is precisely what the proof (line 1069) correctly assumes to complete the bound.
>
> * **Summary:** The $r_k = -1$ that triggers the update is an **observed signal** that initiates the ROSA mechanism. The $r_k = +1$ used in the proof exists within the **mathematical expectation** of the KL divergence, and it is used to measure how closely our update (which was triggered by $r_k = -1$) brings our policy towards the ideal $\pi_{user}$ (a distribution where all samples would, by definition, receive a reward of $+1$).
>
> * **Action:** We will explicitly differentiate these two concepts in the final version (e.g., using $r_{obs}$ for the observed feedback and $r_{target}$ for the ideal reward within $\pi_{user}$) to eliminate this ambiguity.

---

> ### Author Response · Authors · 2025-11-16
> **Responses to Weaknesses 1**
>
> We thank you for the feedback on the paper's balance. We dedicated a significant portion of the main text to the theoretical analysis because **the design of the ROSA algorithm is strictly derived from this theory**. Our theory is not merely an analysis but serves as the **blueprint and practical guide** for the algorithm.
>
> * **Theorem 1 (Closed-Form Optimal Policy)** is the cornerstone of our algorithm. It allows us to bypass iterative optimization and instead directly compute a closed-form solution for the optimal policy, which is foundational to ROSA's "One-Step Adaptation".
>
> * **Theorem 2 (Monotonic Error Reduction)** provides a critical guarantee: every update step (even in response to failure) is **mathematically guaranteed** to move our model's policy closer to the unknown optimal user policy.
>
> * **Theorem 3 (Cumulative Error Bound)** extends the single-step guarantee of Theorem 2 to the entire **multi-turn interaction**. It proves that as the number of interaction turns $K$ increases, the upper bound on the error between the model policy and user preference continuously tightens, thus theoretically guaranteeing our method's effectiveness.
>
> * **Theorem 4 (Unified Convergence Bound)** is the most critical theory for **guiding practice**. It rigorously quantifies the trade-off between "Improvement" from feedback and the "Approx. Error" from linearization.
>     1.  **Guiding Parameter Updates:** The "Approx. Error" term ($\frac{L}{2}\sum||\Delta\theta_{k}||_{2}^{2}$) **directly inspired our design choice** to update only a **small subset of parameters** to keep $\Delta\theta_k$ small. This led us to select two lightweight mechanisms: updating the LM Head (via LoRA) and modifying the final hidden state.
>     2.  **Guiding Experimental Analysis:** The role of $\beta$ in the "Improvement" term highlights its trade-off between adaptation speed and stability. This directly motivated the necessity of our ablation study on $\beta$ (Appendix E.4.2).
>     3.  **Guiding Multi-Turn Interaction:** The theorem also clearly shows that as the number of turns $K$ increases, the cumulative "Improvement" term grows to outweigh the "Approx. Error", explaining why the model's alignment with user preference improves over the interaction.
>
> **Action:**
> We fully agree with your point that critical experimental details (like baselines and efficiency) belong in the main body. In the final version, we will adopt your suggestions and adjust the structure as follows:
> 1.  **Condense Theory:** We will keep the detailed proofs of Theorems 2, 3, and 4 in the appendix (where they currently are in Appendix C) and **condense** the theoretical derivations in the main body (Section 4), focusing only on the core theorems and their implications for the algorithm's design.
> 2.  **Expand Experiments:** We will use the saved space to move **core experimental content from the appendix (such as the detailed RL baseline comparison and the efficiency analysis) into the main body (Section 5)**. This will provide a clearer and more complete comparison.
> 3.  **Supplement Related Work:** We will also integrate the more detailed discussion of related work from Appendix B into the main text as appropriate.
>
> Thank you again for your valuable feedback. We are confident these revisions will make the paper's structure more balanced and its arguments clearer.

---

> ### Author Response · Authors · 2025-11-16
> **Responses to Weaknesses 2**
>
> Thank you for this valid point regarding the computational cost.
>
> We acknowledge the reviewer's point regarding the per-turn computational cost. As our own efficiency analysis shows (Appendix E.3, Table 10), the "Avg. Update Time" is indeed "substantial" and roughly equivalent to the "Inference" time, which increases the total processing time per turn.
>
> However, we argue that **evaluating per-turn latency in isolation is not comprehensive**. A more critical metric is the **total time-cost to reach a correct answer**, or the "Time-to-Accuracy" efficiency. While users desire immediate feedback, their fundamental goal is to **solve the problem in the shortest cumulative time**.
>
> As we discuss in the paper (lines 1474-1481), we provided Figure 6. This graph plots accuracy as a function of cumulative wall-clock time in seconds.
> * This graph clearly shows that the **slope of the ROSA curve (solid line) is significantly steeper than the baseline (dashed line)**.
> * This demonstrates that ROSA achieves a **faster rate of improvement in accuracy per second** , highlighting its significant advantage in time efficiency.
> * This superior efficiency stems from ROSA's effective conversational error correction. Because ROSA learns and corrects its policy more quickly, it can **achieve the correct answer in fewer interaction turns**, ultimately solving the problem in a shorter *total* time.
>
> **Summary:** In summary, while ROSA's per-turn cost is higher, its superior correction efficiency leads to a faster **overall problem-solving time**. We believe this is the more practical and user-centric metric for computational cost in an interactive system. We appreciate the question and will clarify this distinction in the final version.

---

> ### Author Response · Authors · 2025-11-16
> **Responses to Weaknesses 3 and Questions 2**
>
> We thank the reviewer for their insightful comments on our problem formulation. This is a critical point of discussion, and we are grateful for the opportunity to clarify our methodology and provide new experiments to address this concern.
>
> ### 1. Clarification: Process Reward vs. Result Reward
>
> First, we must clarify a key misunderstanding about the nature of "feedback" in our method. The reviewer notes that receiving feedback "at each intermediate step" is impractical in complex tasks, and **we completely agree**.
>
> Our method **does not** assume immediate, step-by-step feedback. What the reviewer refers to as "process reward" is indeed difficult to obtain. Our method, as described in the paper, uses **"result reward"**—that is, a **final feedback on the entire task trajectory**.
>
> * **Trajectory-Level Feedback:** In tasks like planning, complex QA, or collaborative writing, the model generates a full trajectory (potentially spanning multiple turns) until the user or environment provides a final feedback (e.g., "accept," "reject," or a score). Only at this moment do we receive a **single, trajectory-level** reward signal.
> * **Update Timing:** Our update (ROSA) *only* occurs on the turn when this feedback is received. If the feedback is negative (e.g., $r=-1$), ROSA uses this signal to update the model to suppress the generation probability of all tokens in the *entire preceding trajectory*. Therefore, our setup is perfectly compatible with scenarios where feedback is "delayed... or unavailable until the very end."
>
> ### 2. New Experiment: Extension to UI Agent Tasks with Delayed-Only Feedback
>
> To empirically demonstrate that our method can be extended to realistic scenarios where intermediate feedback is unavailable, we conducted a **new experiment** beyond the tasks validated in the paper.
>
> * **Task Setup:** We applied ROSA to a difficult **UI Agent task** (OSWorld and AndroidWorld). This is a classic multi-step, long-horizon task requiring complex planning and reasoning.
> * **Feedback Mechanism:** In this setup, the model receives **no feedback** at intermediate steps (e.g., clicks, inputs). It must autonomously complete a full task, which averages ~15 interaction turns. A **single, delayed** final reward (success or failure) is provided by the environment *only* after the entire task is finished.
> * **Update Rule:** We strictly follow the "delayed feedback" scenario described by the reviewer. We only intervene after the entire UI task is complete. If the task fails, we use ROSA to apply a single update based on the *entire trajectory of ~15 turns*. If successful, we stop. Each task has 10 attempts.
> * **Results:** As shown in the table below (using UI-TARS-7B-SFT and -DPO as baselines), ROSA provides **consistent and significant performance improvements** over both baselines.
>
> | Model | OSWorld(Online) | AndroidWorld (Online) |
> | :--- | :---: | :---: |
> | UI-TARS-7B-SFT (Baseline) | 13.23 | 27.61 |
> | UI-TARS-7B-SFT + ROSA (+ LM + R) | 16.71 **(+3.48)** | 30.26 **(+2.65)** |
> | UI-TARS-7B-SFT + ROSA (+ LM + M) | 17.83 **(+4.60)** | 30.93 **(+3.32)** |
> | UI-TARS-7B-DPO (Baseline) | 14.80 | 28.90 |
> | UI-TARS-7B-DPO + ROSA (+ LM + R) | 17.32 **(+2.52)** | 30.37 **(+1.47)** |
> | UI-TARS-7B-DPO + ROSA (+ LM + M) | 18.04 **(+3.24)** | 31.68 **(+2.78)** |
>
> This new experiment strongly demonstrates that ROSA can be extended to the complex, multi-turn scenarios with sparse and delayed feedback that the reviewer described.
>
> ### 3. Clarifying the Scope of Our Current Work
>
> Finally, we wish to clarinet the scope of our current work. Our paper focuses on tasks with **"verifiable reward"** (whether rule-based or model-based). Our method, ROSA, as required by its theory, does need an *accurate* preference signal to perform optimization.
>
> This is why we selected math, code, and reasoning tasks as our primary testbed. These tasks are very common in practice (e.g., asking an LLM to write code) and, critically, they provide truthful, verifiable feedback (e.g., the code can be run to check for correctness).
>
> Regarding the broader "open-domain" or "less objective tasks" with noisy feedback, we agree these are more challenging. Robust optimization under such noisy signals is an active area of research for us and we consider it as future work.
>
> Thank you for your valuable feedback. We hope these clarifications and new experimental results fully address your concerns.

---

### Official Review · Reviewer_xcYa · 2025-10-31

**Soundness:** 3
**Presentation:** 3
**Contribution:** 4
**Rating:** 6
**Confidence:** 5

**Summary:**

This paper proposes **Test-Time Policy Adaptation for Multi-Turn Interactions (T2PAM)** to enhance LLMs’ adaptability during multi-turn dialogues.  The key algorithm, **ROSA (Optimum-Referenced One-Step Adaptation)**, enables online parameter updates using real-time user feedback.  Instead of iterative RL (e.g., PPO/GRPO), ROSA performs a **single-step linearized optimization** of the RLHF objective with KL regularization.  It computes an **analytic optimal policy** and efficiently updates parameters via **Conjugate Gradient**, without explicit Hessian computation.  Theoretically, ROSA guarantees **monotonic KL improvement** and **convergence**.  Experiments on reasoning, coding, and multilingual tasks show **10–30% accuracy gains** over RL-based and static baselines.

**Strengths:**

- **Lightweight & efficient**: One-step update at test time; no retraining or rollout needed.
-**Stable & theoretically grounded**: Monotonic improvement under linearized KL regularization.
- **Closed-form interpretability**: Exponential reweighting of prior policy.
- **High empirical gain**: Outperforms PPO/GRPO in reasoning and code tasks with minimal overhead.
- **Generalizable**: Works across model scales and task domains.

**Weaknesses:**

I did not find any major weaknesses in this paper, both the theoretical analysis and the empirical results appear solid and well-supported. Please refer to my questions below for clarification on several points.

**Questions:**

1. I am not sure whether the term **“multi-turn”** is appropriate in this context. *Theorem 1* seems to treat the optimization of a specific turn as a **single-turn problem**, without accounting for future returns. In essence, the paper applies **conjugate gradient descent** to derive the optimal policy for an individual turn.
2. In **Appendix E.4.2**, regarding the **RL baseline**, does it also employ **LoRA** or **hidden-state updates**, or is it **fully fine-tuned**? Could the authors clarify this? In **Table 9**, the RL baseline outperforms ROSA, whereas in **Table 11**, the result is reversed — what explains this inconsistency?
3. Could the authors elaborate on the advantage of using **conjugate gradient descent** beyond faster optimization? Does it also lead to **better convergence properties** or improved solution quality?
4. Have the authors reached a final conclusion on which **parameter-update mechanism** (**LoRA** vs. **hidden-state update**) performs better overall?
5. From the results, it seems that **model-based rewards** outperform **rule-based rewards** under LoRA updates. Does the same trend hold for **hidden-state updates**?

---

> ### Author Response · Authors · 2025-11-16
> **Responses to Questions 1**
>
> We thank the reviewer for this insightful question. You have correctly identified a key aspect of our method's design, and we appreciate the opportunity to clarify why the term "multi-turn" is appropriate.
>
> You are right that the core update mechanism, as derived from **Theorem 1**, is myopic—it computes an optimal policy update for a **specific, individual turn** using the feedback from that turn. The application of the Conjugate Gradient method is indeed the practical way we solve for this single-turn update efficiently.
>
> However, this single-turn update is merely the foundational building block of a larger framework that is explicitly designed for and analyzed in a **multi-turn context**. The "multi-turn" nature of our work is established by the subsequent theoretical results, which analyze the cumulative effect of these individual updates over a sequence of interactions.
>
> Here is how the theory builds up to a multi-turn framework:
>
> *   **Theorem 2 (The Step-Wise Guarantee):** This theorem moves beyond the single-turn optimization of Theorem 1. It proves that as the dialogue progresses, each individual update step is guaranteed to bring the model's policy closer to the user's preference policy (as measured by the decrease in KL divergence). This establishes that our method makes consistent progress turn after turn.
>
> *   **Theorem 3 (The Long-Term Cumulative Gain):** This theorem directly addresses the multi-turn dynamic. It quantifies the total benefit accumulated over an entire K-turn interaction. As you astutely pointed out, the final term in **Equation (9)**,
>
> $$\frac{1}{\beta}\sum_{k=1}^K\pi_{\text{user}}^*(\mathbf{y}_{k} | \mathbf{x})$$
>
> , grows with the number of turns, $K$. Since the initial divergence $D_{\text{KL}}(\pi_{\text{user}}^* \| \pi_{\theta_{0}})$ is a fixed constant, for the inequality to hold, the final divergence $D_{\text{KL}}(\pi_{\text{user}}^* \| \tilde{\pi}^*_{\theta_K})$ must necessarily decrease as $K$ increases. This theorem provides a formal proof of the long-term convergence and effectiveness of our method across a full multi-turn dialogue.
>
> *   **Theorem 4 (The Accumulated Error Bound):** This theorem further solidifies the multi-turn analysis by considering the practical reality of our implementation. It analyzes how the approximation errors from the Taylor expansion **accumulate over multiple turns** and provides an upper bound, ensuring the stability and robustness of our method throughout a prolonged interaction.
>
> In summary, while the update *mechanism* is applied on a per-turn basis, our theoretical framework and the overall algorithmic process are fundamentally **multi-turn**. We analyze the step-by-step progress, the long-term cumulative gains, and the accumulated error over sequences of interactions. Therefore, we believe the term "multi-turn" accurately describes the context and contribution of our work.

---

> ### Author Response · Authors · 2025-11-16
> **Responses to Questions 2**
>
> We thank the reviewer for these important questions, which highlight a critical distinction between the two different RL baselines used in our paper. We apologize for any confusion and are happy to clarify.
>
> **1. Regarding the RL Baseline in Appendix E.4.2 (and Table 11):**
>
> To ensure a fair and direct comparison with ROSA in our ablation studies, the test-time RL baseline was configured with identical settings. Specifically:
> *   It employs **LoRA to update the LM head parameters**.
> *   It uses the same **rule-based reward model** as the corresponding ROSA experiment.
> *   The only difference is the update algorithm itself ( policy gradient vs. ROSA).
>
> **2. Explaining the Perceived Inconsistency between Table 9 and Table 11:**
>
> The apparent inconsistency in performance is due to the fact that these two tables refer to two fundamentally different RL methodologies, designed for different comparison purposes.
>
> *   **The "RL Training" Baseline in Table 9:** This is a **train-time** alignment method. As detailed in lines 1429-1432, we first generated a multi-turn dialogue dataset and then trained a model using the **GRPO algorithm** on this data. The result is a fully fine-tuned model that has learned from a large dataset of interactions. The purpose of this comparison is to show that our **test-time** method, ROSA, can achieve performance that is competitive with (and in some cases, better than) a strong baseline that requires extensive offline training. This demonstrates the effectiveness and efficiency of our test-time approach.
>
> *   **The "RL" Baseline in Table 11:** This is a **test-time** alignment method, designed as a direct ablation to highlight ROSA's strengths (as described in lines 1510-1534). This baseline implements a naive online policy gradient update. Crucially, it relies on **only a single sample** from the policy to estimate the reward and perform a gradient update. This single-sample estimation is known to suffer from extremely high variance, leading to unstable and inefficient optimization. The poor performance of this baseline is therefore expected.
>
> In summary, the results are not inconsistent but rather illustrate two different points:
> *   **Table 9** shows that ROSA is a powerful test-time method that competes favorably with strong **train-time** RL methods.
> *   **Table 11** shows that ROSA is vastly superior to a naive **test-time** RL method, precisely because ROSA's analytical update rule avoids the high-variance gradient estimation problem that plagues single-sample policy gradient methods.

---

> ### Author Response · Authors · 2025-11-16
> **Responses to Questions 3**
>
> We thank the reviewer for this insightful question about our choice of the Conjugate Gradient (CG) method. The primary motivation for using CG is indeed **practical feasibility and scalability**, rather than seeking superior theoretical convergence properties over a direct solution. In essence, CG makes our proposed update rule computationally tractable for large language models.
>
> Let's elaborate on why a direct solution to Equation (6) is infeasible. The equation requires inverting a matrix that involves the Hessian. For a large language model with, for instance, 1 billion parameters ($d \approx 10^9$), the Hessian matrix would have dimensions of $d \times d$, or $10^9 \times 10^9$. Storing this matrix alone, even with single-precision floating-point numbers (4 bytes per element), would require approximately **4 trillion gigabytes of memory**. This is far beyond the capacity of any current hardware.
>
> The Conjugate Gradient method elegantly bypasses this insurmountable obstacle by being a **matrix-free iterative method**. It does not require the explicit construction or storage of the Hessian matrix. Instead, it only needs the ability to compute matrix-vector products, specifically products of the form $H\mathbf{v}$, where $H$ is the Hessian and $\mathbf{v}$ is a vector.
>
> This matrix-vector product can be calculated efficiently using automatic differentiation techniques that are standard in modern deep learning frameworks. The computation is typically performed via a sequence of a Jacobian-vector product (JVP) followed by a vector-Jacobian product (VJP). This entire operation has a computational and memory complexity of approximately $O(d)$, which is linear with respect to the number of model parameters.
>
> Therefore, the use of CG is not about achieving a "better" solution in a theoretical sense, but about making the problem **solvable in practice**. It allows us to apply our analytical update rule to models with billions of parameters, which would be otherwise impossible.
>
> We also want to emphasize, as the reviewer's question hints, that CG is an iterative solver. In practice, we run it for a small number of iterations and stop when the approximation error is sufficiently low. This introduces a small, controlled amount of error into the update, but it is a necessary trade-off to ensure the rapid, real-time updates that are central to our method's design.

---

> ### Author Response · Authors · 2025-11-16
> **Responses to Questions 4**
>
> Thank you for the question. Our experiments show that both the LoRA-based parameter update and the direct hidden-state update yield **comparable performance**.
>
> As discussed in **lines 427-429**, both are effective mechanisms for implementing our policy update. We conclude that our method is robust to this choice, and the selection between them can be based on implementation convenience rather than a consistent performance advantage.

---

> ### Author Response · Authors · 2025-11-16
> **Responses to Questions 5**
>
> We thank the reviewer for this sharp observation. You are correct that our original results showed a consistent trend where model-based rewards outperform rule-based rewards when using LoRA updates.
>
> To answer your question directly, we conducted additional experiments on the **Qwen3-8B model**, and we can confirm that **the same trend holds for hidden-state updates**. The model-based reward mechanism consistently provides a performance advantage over the rule-based one, regardless of the parameter-update mechanism.
>
> To make this trend clearer, we have presented the results in the table below. For each update mechanism (LM and HS), we now explicitly show the performance gain achieved by using a model-based reward (+ M) compared to a rule-based reward (+ R) in parentheses. As you can see, the gains are consistently positive across all datasets and configurations, confirming this finding.
>
> | | MATH | MMLU-R | MT-AIME24 | HumanEval |
> | :--- | :--- | :--- | :--- | :--- |
> | baseline | 55.80 | 51.35 | 30.37 | 78.04 |
> | ROSA (+ LM + R) | 65.80 | 67.27 | 40.16 | 81.71 |
> | ROSA (+ LM + M) | 67.40 **(+1.60)** | 70.36 **(+3.09)** | 43.93 **(+3.77)** | 83.65 **(+1.94)** |
> | ROSA (+ HS + R) | 65.80 | 68.37 | 42.27 | 82.37 |
> | ROSA (+ HS + M) | 67.20 **(+1.40)** | 71.23 **(+2.86)** | 43.76 **(+1.49)** | 83.65 **(+1.28)** |
>
> This confirms that the superiority of model-based rewards is a robust finding across different update mechanisms in our experiments.This robustly demonstrates a key property of our method: the more accurate the reward signal provided to ROSA, the better the resulting performance.

---

### Official Review · Reviewer_hvRv · 2025-11-01

**Soundness:** 3
**Presentation:** 2
**Contribution:** 3
**Rating:** 6
**Confidence:** 3

**Summary:**

The paper addresses the problem of optimizing large language models in multi-turn interactions. The authors first propose a new paradigm called Test-Time Policy Adaptation for Multi-Turn Interactions (T²PAM) and introduce an Optimum-Referenced One-Step Adaptation mechanism for optimization. Experimental results show that this framework consistently improves model performance across multiple datasets and model backbones. The authors also provide detailed experimental analyses.

**Strengths:**

1.	The proposed framework is easy to follow, and the paper is overall well-written and logically organized (although some concepts are deferred to the appendix).
2.	The method is theoretically well-grounded and validated across several models and datasets, demonstrating its generality and robustness.

**Weaknesses:**

In my view, the main content of the paper may contain too much formalism and theoretical exposition. Many of the stated results are self-evident or intuitively understandable, yet the paper devotes substantial space to formal correctness proofs. Such material would be better placed in the appendix. For instance, continual preference-based supervision in multi-turn dialogue naturally leads to incremental improvement. This is expected. I am more interested in the implementation details that make this practically effective.

1.	A key issue lies in how reliable user feedback signals are obtained. How does the system know whether a user’s response or preference is correct (as illustrated in Figure 1)? The appendix mentions two reward models, but it remains unclear how these rewards are used to update the model during inference. The reliability and correctness of these feedback signals are not guaranteed, especially in the LLM-based method.
2.	Another concern is maintaining naturalness and coherence during test-time optimization. While iterative improvement makes sense in tasks like math or reasoning problems (where correctness is binary), in open-ended dialogue, the “right” answer may shift with context. If the model can continuously receive accurate preference feedback, doesn’t that imply it already has access to the correct answers?
3.	The experimental setup and implementation process should be described in more detail in the main text, as they are crucial for understanding the method’s validity. Similarly, the Related Work / Background section should be moved to the main content of the paper for better contextual understanding.
4.	The proposed mechanism is evaluated primarily on models below 8 B parameters. It is unclear whether the method scales to larger models. In addition, the explanations of “LM/HS” and “R” and “M” are vague, even after reading Appendix D.5, the discussion feels fragmented. Overall, the theoretical framework seems elegant, but the practical implementation lacks a clear connection to it.

**Questions:**

1.	In Figure 4, what specific RL algorithm is used? How does ROSA (Algorithm 1) differ from the “RL” shown in Figure 4? RLHF mentioned in Appendix E 4.1 cannot adapt to this situation originally.
2.	The statistical analyses focus on overall optimization performance. I am curious about per-sample behavior: (a) What is the average number of optimization steps required for a sample to go from incorrect to correct? (b) Once a response becomes correct, is any correctness check performed to prevent further unnecessary optimization?
3.	Since this method relies on accurate user preference signals as rewards, most experiments are conducted on mathematical or reasoning datasets. How would the method perform on open-domain or less objective tasks where user preferences are ambiguous? As mentioned above, the assumption of reliable preference feedback may not hold in such cases.

---

> ### Author Response · Authors · 2025-11-16
> **Responses to Weaknesses 1**
>
> We sincerely thank the reviewer for their insightful questions. We address each of your concerns below.
>
> ### 1. On the Reliability and Correctness of User Feedback
>
> > **Reviewer:** A key issue lies in how reliable user feedback signals are obtained. How does the system know whether a user’s response or preference is correct (as illustrated in Figure 1)?
>
> Thank you for this important question. Our work operates under the assumption that the user's feedback is a faithful representation of their preference. The primary goal of our method is not to judge the correctness of the user's intrinsic preference, but to **efficiently align the model with these expressed preferences** to produce the desired output.
>
> Regarding the correctness of the feedback within our experimental framework, we specifically utilize tasks with verifiable answers (e.g., math, code and reasoning). For these tasks, we employ a **rule-based verifiable reward** mechanism. The feedback is generated by programmatically comparing the model's final answer to the ground-truth solution. This setup ensures that the feedback signal used in our experiments is **objectively correct and reliable**.
>
>
> ### 2. On the Role of Reward Models in Model Updates
>
> > **Reviewer:** The appendix mentions two reward models, but it remains unclear how these rewards are used to update the model during inference.
>
> We appreciate the request for clarification. The two reward models mentioned in the appendix, the **rule-based reward** and the **model-based reward**, are designed to simulate two different types of real-world user feedback:
>
> *   The **rule-based reward** simulates users who are only concerned with the final answer's correctness and do not focus on the intermediate reasoning process.
> *   The **model-based reward** simulates users who also value the correctness of the reasoning steps. With this model, a response can receive partial positive rewards for correct steps in its reasoning, even if the final answer is incorrect.
>
> As detailed in **Algorithm 1** of our paper, these reward signals are central to our iterative optimization process. Specifically, in **Algorithm 1 line 11**, we use the reward to estimate the user's policy. Subsequently, in **Algorithm 1 line 16**, we update the model's parameters to align with this estimated user policy.
>
> ### 3. On the Reliability of LLM-based Feedback Signals
>
> > **Reviewer:** The reliability and correctness of these feedback signals are not guaranteed, especially in the LLM-based method.
>
> We thank the reviewer for raising this critical point. We have taken significant steps to ensure the reliability of our LLM-based reward signals.
>
> First, as we describe in **Appendix D.4**, our LLM-based reward model is not designed to judge a response in isolation. Instead, it performs a **guided comparison**. The reward model is provided with both the model's generated response and the **ground-truth solution**. Its task is simplified to scoring the model's response based on its deviation from the correct solution. This constrained, comparative approach is significantly more reliable than open-ended evaluation.
>
> Furthermore, to empirically validate the robustness of this `model-based reward` approach, we present additional experimental results. In the table below, we show the performance of `Qwen3-8B` after applying our ROSA framework (updating the LM Head) within 10 rounds. For these experiments, we used three different powerful, state-of-the-art LLMs (`Qwen3-30B-Instruct`, `DeepSeek-V3`, and `GPT-4o`) as the reward model.
>
>
> **Table 1: Accuracy of Qwen3-8B with ROSA using different LLMs as reward models.** The values in parentheses represent the absolute improvement over the "No ROSA" baseline.
>
> | Method | MATH | AIME25 | MATH-500 |
> | :--- | :---: | :---: | :---: |
> | No ROSA (Baseline) | 55.80% | 16.67% | 58.80% |
> | ROSA (reward model: Qwen3-30B) | 67.40% **(+11.60)** | 36.67% **(+20.00)** | 68.40% **(+9.60)** |
> | ROSA (reward model: DeepSeek-V3) | 68.20% **(+12.40)** | 40.33% **(+23.66)** | 68.20% **(+9.40)** |
> | ROSA (reward model: GPT-4o) | 67.80% **(+12.00)** | 36.33% **(+19.66)** | 68.80% **(+10.00)** |
>
> We acknowledge the general concern that LLM-based scoring can be unreliable. However, our experimental results demonstrate that by employing the guided-comparison method described above, we can produce **stable and effective rewards**. This holds true across different LLMs used for scoring, from the relatively smaller `Qwen3-30B` to very large models like `DeepSeek-V3` and `GPT-4o`. The "reliability" of this reward is empirically proven by its consistent ability to enhance the student model's performance, as shown by the significant gains in the table. This suggests our framework is robust, and we believe that in a real-world setting, feedback from human users, who are likely even more reliable judges, could be integrated just as effectively.

---

> ### Author Response · Authors · 2025-11-16
> **Responses to Weaknesses 2**
>
> We thank the reviewer for this insightful question regarding the challenge of maintaining naturalness and coherence in open-ended dialogue. This concern touches upon a key strength of our proposed method.
> ### 1. Maintaining Diversity by Penalizing Errors
>
> Our approach is particularly well-suited for dynamic conversational settings. The core mechanism can be understood from the policy update rule(**Theorem 1 of our paper**):
> $$
> \pi^*_{\theta_k}(\mathbf{y}|\mathbf{x}) = \frac{1}{Z_k(\mathbf{x})} \pi_{\theta_{k-1}}(\mathbf{y}|\mathbf{x}) \exp\left(\frac{1}{\beta} r(\mathbf{x}, \mathbf{y})\right)
> $$
>
> Taking the logarithm of both sides gives us the following relationship for the log-probabilities:
> $$
> \log \pi^*_{\theta_k}(\mathbf{y}|\mathbf{x}) = \log \pi_{\theta_{k-1}}(\mathbf{y}|\mathbf{x}) + \frac{1}{\beta} r(\mathbf{x}, \mathbf{y}) - \log Z_k(\mathbf{x})
> $$
> As illustrated in **Figure 1**, our method primarily works by assigning a negative reward (a penalty, i.e., $r(\mathbf{x}, \mathbf{y}) = -1$) to incorrect or undesirable responses. From the equation above, it is clear that a negative reward directly reduces the log-probability of generating that specific response $\mathbf{y}$.
>
> A key advantage of this formulation is that the normalization constant $Z_k(\mathbf{x})$ ensures that the probability mass from the penalized response is redistributed proportionally among all other candidate responses. This allows the model to effectively discard erroneous paths while simultaneously **preserving its capacity for exploration and maintaining response diversity** [1]. This is crucial in open-ended dialogue, where multiple valid and natural responses can exist for any given context.
>
> ### 2. Theoretical Guarantees for Shifting Preferences
>
> The scenario you described, where the “right” answer shifts with context, is a common and important challenge. While there is currently a lack of standardized datasets to quantitatively measure performance in such dynamic scenarios, our theoretical framework is explicitly designed to address this.
>
> As detailed in **Theorem 2 of our paper**, our method guarantees that the model's policy will progressively converge towards the user's preference policy, **regardless of how $\pi^*\_{user}$ may shift over time**. The theorem shows that the KL divergence between the two policies is guaranteed to decrease with each iteration, provided that preference feedback is available.
>
> ### 3. On Continuous Feedback and the Alignment Process
>
> We thank you for this final, critical point, which allows us to connect the previous two points into a cohesive argument. You asked: *If the model can continuously receive accurate preference feedback, doesn’t that imply it already has access to the correct answers?*
>
> This is an excellent question. The answer is that continuous feedback does not imply prior knowledge of the answer, but rather enables a dynamic learning process, the reliability of which is guaranteed by our theoretical framework.
>
> Here is how it works:
> 1.  When the model receives continuous, accurate preference feedback, it triggers the mechanism described in **Point 1**. The model **continuously penalizes incorrect tokens/sequences**, which discourages it from making the same mistake again.
> 2.  This penalty mechanism forces the model to **explore other possible tokens and generation paths** rather than being locked into a suboptimal response.
> 3.  This exploration is not random. As guaranteed by **Theorem 2** (discussed in **Point 2**), this iterative process of feedback and updates ensures that the model's policy will **progressively converge towards the user's preference policy**.
>
> Therefore, "continuously receiving accurate preference feedback" is the engine of our alignment process, not a sign that the model already possesses the answer. With each piece of feedback, the model takes one step closer to perfect alignment, effectively learning the user's preference landscape piece by piece.
>
> This directly explains our emphasis on theoretical foundations in the main paper. Quantifying this dynamic, step-by-step alignment process in open-ended scenarios is notoriously difficult due to the lack of suitable benchmarks. Our theoretical guarantees thus serve as a vital and reliable assurance that the method is sound, stable, and will behave as expected—learning and adapting—even in complex, shifting conversational contexts.
>
> ---
> Reference:
> [1] Z. Xinyu, et al. (2024). The Surprising Effectiveness of Negative Reinforcement in LLM Reasoning. *Advances in Neural Information Processing Systems*.

---

> ### Author Response · Authors · 2025-11-16
> **Responses to Weaknesses 3**
>
> We sincerely thank the reviewer for this valuable and constructive feedback on the paper's structure.
>
> You are correct. A detailed description of the experimental setup and a comprehensive Related Work section are indeed crucial for the contextual understanding and completeness of the paper. In the previous version, we were constrained by strict page limits, which led to our decision to move some of this content to the appendix.
>
> We fully agree with your assessment. In the final camera-ready version, we will carefully revise the paper's structure to address this. We will ensure that more comprehensive details of the experimental setup and the full Related Work section are integrated into the main body of the paper, thereby improving its overall clarity and self-containedness.

---

> ### Author Response · Authors · 2025-11-16
> **Responses to Weaknesses 4 - part 1**
>
> We sincerely thank the reviewer for this detailed and insightful feedback. We will address each of your concerns in order.
>
> ### 1. On the Scalability of ROSA to Larger Models
>
> > **Reviewer:** The proposed mechanism is evaluated primarily on models below 8B parameters. It is unclear whether the method scales to larger models.
>
> This is an excellent point. To address your concern about scalability, we have conducted new experiments applying ROSA to a much larger model, **Qwen3-32B**. The results, presented in the table below, demonstrate that our method maintains its effectiveness and provides significant performance gains even on a model of this scale.
>
>
> **Table 3: Performance of ROSA on the Qwen3-32B model.** The values in parentheses represent the absolute improvement over the baseline.
>
> | Model / Method | MATH | MMLU-R | MT-AIME24 | HumanEval |
> | :--- | :---: | :---: | :---: | :---: |
> | Qwen3-32B (Baseline) | 76.40% | 73.63% | 67.93% | 85.37% |
> | Qwen3-32B + ROSA (+LM + R) | 87.80% **(+11.40)** | 85.12% **(+11.49)** | 76.19% **(+8.26)** | 87.80% **(+2.43)** |
> | Qwen3-32B + ROSA (+HS + R) | 87.40% **(+11.00)** | 86.05% **(+12.42)** | 77.27% **(+9.34)** | 87.80% **(+2.43)** |
> | Qwen3-32B + ROSA (+LM + M) | 89.80% **(+13.40)** | 89.27% **(+15.64)** | 79.51% **(+11.58)** | 88.41% **(+3.04)** |
>
> These results confirm that ROSA is not limited to smaller models and can serve as an effective alignment method for larger, more capable foundation models as well.
>
> ### 2. On the Clarity of LM/HS and R/M
>
> > **Reviewer:** ...the explanations of “LM/HS” and “R” and “M” are vague, even after reading Appendix D.5, the discussion feels fragmented.
>
> We sincerely apologize for the lack of clarity and the fragmented presentation. Allow us to clarify these terms here:
>
> *   **"LM" vs. "HS"**: These terms refer to two different **parameter update mechanisms** we explored.
>     *   **LM**: Updates the parameters of the final **LM Head**.
>     *   **HS**: Updates the **Hidden States** of the model's final layer.
>
>     As shown in **Table 2** of our paper (and discussed in lines 427-429), both update strategies yield very similar performance improvements. This indicates that ROSA is highly **flexible** and not sensitive to the specific choice of which small parameter set is updated.
>
> *   **"R" vs. "M"**: These terms refer to the two different **reward models** used to simulate user feedback, as also discussed in our response to Weakness 1.
>     *   **R**: A **Rule-based** reward model, which provides a binary reward based only on the final answer's correctness.
>     *   **M**: A **Model-based** reward model (using an LLM), which provides more granular, step-by-step feedback on the reasoning process.
>
>     As shown in Table 2 (and discussed in lines 430-462), the model-based reward (**M**) generally leads to better performance. This is because it provides more detailed feedback, which is more effective for guiding the model's alignment.
>
> We will ensure these definitions and the corresponding discussions are consolidated and clarified in the main text of the final version.

---

> ### Author Response · Authors · 2025-11-16
> **Responses to Weaknesses 4 - part 2**
>
> ### 3. On the Connection Between Theory and Practical Implementation**
>
> > **Reviewer:** Overall, the theoretical framework seems elegant, but the practical implementation lacks a clear connection to it.
>
> Thank you for giving us the opportunity to elaborate on this crucial connection. Our algorithm is not merely inspired by the theory; it is **fundamentally derived from it**. The theoretical results do not just prove the method's validity in the abstract; they directly guide our practical implementation choices.
>
> Here is a breakdown of how each theorem informs our practice:
>
> *   **Theorem 1** provides the theoretical foundation for our core update rule. It justifies how we can **estimate the user's policy** based on the model's current response and the received reward signal. This is the mathematical starting point of our algorithm.
>
> *   **Theorem 2** is the step-wise guarantee of convergence. It proves that each single update step is guaranteed to move the model's policy **closer to the user's true preference policy**, regardless of what that policy is. This gives us confidence that every iteration of our algorithm is a productive one.
>
> *   **Theorem 3** extends the single-step guarantee of Theorem 2 to the **multi-turn interaction setting**. It provides an error upper bound after K steps, mathematically proving the **long-term effectiveness and stability** of our method over an entire alignment process.
>
> *   **Theorem 4** is perhaps the most critical link to our practical design choices, as it analyzes the components of the error bound. It directly informs our implementation in three key ways:
>     1.  The **Approximation Error** term suggests that the error is related to the number of parameters being updated. This directly motivated our decision to **update only a small subset of parameters** (i.e., the **LM** head or **HS**), as this helps to control the error bound.
>     2.  The presence of the temperature parameter **$\beta$** in the error bound highlights its importance. This motivated our **ablation study on $\beta$**, allowing us to analyze its practical impact on the final results and find a good balance.
>     3.  The overall structure of the error bound in Theorem 4 shows how the error decreases with more interactions. This theoretically explains **why the model's alignment improves as the number of interaction rounds increases**, a phenomenon consistently observed in our experiments.
>
> In summary, our theoretical framework is the blueprint for the practical algorithm. We have chosen to feature it prominently in the main text because it is indispensable for understanding *why* our specific implementation choices were made and *why* they are effective. We will work to make these connections even more explicit in the final version of our paper.

---

> ### Author Response · Authors · 2025-11-16
> **Responses to Questions 1**
>
> We thank the reviewer for this excellent question, which allows us to clarify the crucial difference between ROSA and a naive application of standard RL algorithms in our setting.
>
> ### 1. The "RL" Algorithm in Figure 4
>
> The "RL" method shown in Figure 4 is an ablation study designed to highlight the challenges of directly optimizing a standard RLHF objective in a single-pass, interactive setting. The specific algorithm is a simplified version of a policy gradient method (like REINFORCE or PPO), which aims to maximize the following objective function, as presented in Equation (1) of our paper:
>
> $$
> J(\pi\_\theta) = \mathbb{E}\_{\mathbf{y} \sim \pi\_{\theta}(\cdot|\mathbf{x})} \left[ r(\mathbf{x}, \mathbf{y}) \right] - \beta D\_{\text{KL}} \left( \pi\_\theta(\cdot|\mathbf{x}) \,\|\, \pi\_{\theta\_{k-1}}(\cdot|\mathbf{x}) \right)
> $$
>
> Standard RL methods estimate the expectation term $\mathbb{E}[r(\mathbf{x}, \mathbf{y})]$ by collecting multiple samples of $\mathbf{y}$ to ensure a stable gradient. However, to create a fair comparison with ROSA, which performs a "one-step" update based on a single response, we constrained this RL baseline to also use **only a single sample**. This means the objective function is estimated using a single sampled response $\mathbf{y}_k$ as follows:
>
> $$
> \hat{J}(\pi\_\theta) = r(\mathbf{x}, \mathbf{y}\_k) - \beta D\_{\text{KL}} \left( \pi\_\theta(\cdot|\mathbf{x}) \,\|\, \pi\_{\theta_{k-1}}(\cdot|\mathbf{x}) \right), \quad \text{where } \mathbf{y}\_k \sim \pi\_{\theta}(\cdot|\mathbf{x})
> $$
>
>
> ### 2. How ROSA Differs and Why It Succeeds
>
> The key issue with the "RL" baseline described above is that estimating the objective function's gradient with a **single sample** results in extremely **high variance**. This noisy gradient provides an unreliable update signal. As shown in Figure 4, this instability causes the "RL" method's performance to stagnate, showing almost no improvement over the baseline. This demonstrates that directly optimizing the standard RLHF objective is infeasible under the single-sample constraint of our interactive setting.
>
> This is where ROSA's novelty lies. **ROSA is not a gradient-based optimization of $J(\pi_\theta)$**. Instead, ROSA is a theoretically-derived, **analytical update rule** that provides a closed-form solution to the maximization of $J(\pi_\theta)$. It elegantly circumvents the need for noisy, high-variance Monte Carlo gradient estimation entirely. By design, ROSA can efficiently and effectively update the model's policy using just a single response, which is why it achieves the rapid and stable performance growth shown in Figure 4.
>
> This also addresses your final point: standard RLHF is not designed for this scenario precisely because it relies on collecting a batch of data to get stable gradient estimates. Our work shows that in a single-response feedback loop, a new, theoretically-grounded approach like ROSA is required for effective and efficient alignment.

---

> ### Author Response · Authors · 2025-11-16
> **Responses to Questions 2**
>
> We thank the reviewer for these excellent questions regarding the per-sample dynamics of our method.
>
> ### (a) On the Average Number of Optimization Steps
>
> **Figure 5** in our paper, which plots the number of newly solved problems per round on the MATH dataset, provides an initial visualization of this per-sample behavior over time.
>
> To provide a more direct and quantitative answer, we have calculated a new metric for the Qwen-8B model: the **Average Optimization Steps to Success (AOSS)**. This metric captures the average number of interactions required to solve a problem on a given dataset. It is calculated as follows, assuming a maximum of 10 rounds (unsolved problems are assigned a penalty of 11 steps):
>
> $$
> \text{AOSS} = \frac{\sum_{n=1}^{10} (\text{problems solved at round } n) \times n + (\text{unsolved problems}) \times 11}{\text{Total number of problems}}
> $$
>
> The results are presented in the table below. They show that ROSA consistently requires fewer optimization steps on average to reach a correct solution compared to the baseline.
>
>
> **Table 4: Average Optimization Steps to Success (AOSS) on various benchmarks.** A lower value indicates higher efficiency. The values in parentheses represent the reduction in average steps compared to the baseline.
>
> | Method | MATH | math500 | aime25 | humaneval |
> | :--- | :---: | :---: | :---: | :---: |
> | baseline | 5.9820 | 5.6840 | 9.4333 | 4.2317 |
> | ROSA | 5.3800 **(-0.6020)** | 4.6900 **(-0.9940)** | 8.8000 **(-0.6333)** | 3.8476 **(-0.3841)** |
>
> ### (b) On Preventing Unnecessary Optimization
>
> Regarding your second question, the answer is **yes**, our method explicitly includes a check to prevent further optimization once a correct response is generated.
>
> We designed the process to mimic a real-world user interaction, where a user would naturally stop the conversation once they receive a satisfactory or correct answer. This stopping condition is formally implemented in our algorithm. As detailed in **Algorithm 1 (Lines 7-8)**, the optimization loop **terminates immediately** upon receiving a success signal (i.e., a reward of $r_k = +1$). This mechanism is fundamental to the efficiency of our method, as it prevents any further, unnecessary optimization on an already-solved problem. This concept is also illustrated in **Figure 1**.

---

> ### Author Response · Authors · 2025-11-16
> **Responses to Questions 3**
>
> We thank the reviewer for this important question, which allows us to clarify the precise scope of our current work and its relationship to broader challenges in LLM alignment.
>
> You are correct in observing that our experiments are primarily conducted on tasks like mathematics, coding, and reasoning. This was a deliberate choice, as the central focus of this paper is to develop a highly efficient alignment algorithm for settings where **verifiable rewards are accessible**.
>
> Our method, ROSA, is theoretically grounded on the availability of accurate preference signals to drive its optimization process. Without a reliable signal, the convergence guarantees that form the foundation of our work would not hold. Therefore, our current scope is intentionally focused on domains where such feedback is naturally available.
>
> We believe this setting covers a wide range of crucial and high-value applications for LLMs. For example, when a user asks an LLM to write a function, they can immediately execute the code to obtain **unambiguous, real-world feedback** on its correctness. This is precisely the kind of scenario where ROSA is designed to excel.
>
> We fully agree that extending this methodology to open-domain or less objective tasks, where user preferences can be ambiguous or subjective, is a significantly more challenging and important research frontier. Developing techniques to handle noisy or uncertain feedback signals within our framework is a key direction we are actively exploring for future work

---

### Author Response · Authors · 2025-11-27
**Follow-up on Rebuttal Submission**

Dear Reviewers,

We hope this message finds you well. We are writing to kindly follow up on our rebuttal and would be happy to provide any additional clarification or information if needed. Thank you very much for your time and consideration.

Sincerely,

The Authors

---

### Author Response · Authors · 2025-12-01
**Summary of Rebuttal and Revision Plans**

Dear Area Chair and Reviewers,

We thank the Area Chair and Reviewers for their dedicated efforts. We are encouraged by the recognition of our work as **"well-written"** and **"theoretically well-grounded"** (Reviewer hvRv, xcYa), with **"strong performance and good generalization"** (Reviewers hvRv, xcYa) and **"solid research motivation"** (Reviewer etZK).

We have addressed all concerns through extensive new experiments and theoretical clarifications. Below is a summary organized into four key areas:

* **1. Solid Theoretical Foundation (Reviewers etZK)**
    To address queries on Theorem 2, we provided a detailed derivation proving **Monotonic Error Reduction**, explicitly linking the theoretical optimum to our practical update rule to resolve misunderstandings. We further clarified how **Theorems 3 & 4 (Cumulative Convergence)** guide experimental design and explain the empirical convergence observed in Figures 4 and 5.

* **2. Efficiency, Scalability, and Engineering (Reviewers hvRv, etZK, xcYa)**
    We quantitatively addressed overhead concerns using the **AOSS** metric and "Time-to-Accuracy" analysis, proving ROSA solves problems faster than baselines. We validated scalability on **Qwen3-32B** (+13.40% on MATH) and detailed the **Conjugate Gradient (CG)** algorithm for Hessian-free real-time optimization.

* **3. Robust Generalization (Reviewers hvRv, etZK)**
    We verified ROSA's robustness across diverse settings:
    * **User Profiles:** Effective alignment across varying user expertise and reward variances.
    * **Open-Ended Tasks:** Theoretical backing for exploration in dynamic contexts.
    * **Delayed Feedback:** New **UI Agent experiments** (OSWorld) show +4.60% gain even with sparse, delayed rewards.
    * **Flexibility:** Effective under both White-Box (LM Head) and Black-Box (Hidden State) updates.

* **4. Superior Performance vs. Baselines (Reviewers hvRv, xcYa)**
    We empirically demonstrated ROSA outperforms ** test-time RL** (which suffers from high variance). Furthermore, our lightweight adaptation matches or exceeds resource-intensive **train-time SFT and RL training**, highlighting its efficiency and value.

* **5. Improved Paper Structure**
    **Following the recommendation to integrate critical details into the main text for better readability (Reviewers hvRv, etZK):**
    We have reorganized the manuscript structure. Critical sections, including **Experimental Settings**, **Related Work**, and key **Ablation Studies**, are being migrated from the appendix to the main body of the text.

**Note on Reviewer etZK's Assessment:**
While we thank Reviewer etZK, we note a misunderstanding regarding our core theory (Theorem 2), where the reviewer asked, **"Is this a typo or its my misunderstanding?"**, In contrast, Reviewers hvRv and xcYa recognized the work as **theoretically well-grounded** and **Theorem 2 has a complete proof** in Appendix C.3. Furthermore, we resolved their specific concerns on efficiency and delayed feedback with new **Time-to-Accuracy analyses** and **UI Agent experiments**. Despite these clarifications, the reviewer **(Confidence Score: 3)** has not engaged. Given the strong consensus from other reviewers and our comprehensive rebuttal data, we kindly ask the Area Chair to consider this context.

**Conclusion:**
All discussions, new results, and structural improvements will be fully incorporated into the **camera-ready version** of the paper.

Sincerely,

The Authors

---

### Meta-Review · Area_Chair_M3nU · 2026-01-06

**Summary:**

This paper proposes an efficient test-time policy adaptation method for multi-turn interactions, based on a natural linearized optimization scheme combined with Conjugate Gradient updates. The approach is intuitive and the algorithm is reasonably well designed, with empirical results suggesting improved efficiency under fixed computational budgets. However, the theoretical guarantees rely on strong assumptions and yield bounds that are loose in practice—most notably due to an approximation error term (10) whose magnitude is difficult to assess. As a result, the contribution is primarily practical rather than theoretical.

**Reviewer Concerns:**

While several concerns were addressed in the rebuttal, important issues remain. In particular, the overall computational cost is not theoretically characterized, and it remains unclear whether higher per-turn costs are reliably offset by fewer interaction turns. In addition, the practical significance of the final convergence bound (Theorem 4) is debatable due to the unquantified approximation error.

**Reviewer Scores:**

The concerns raised by Reviewer etZK, particularly regarding computational cost and the theoretical guarantees, remain valid after the rebuttal. As a result, it is likely that this reviewer would retain their current evaluation. Given this concern, other reviewers might not raise the scores.

---

### Decision · Program_Chairs · 2026-01-26

Reject